# REPAIR: Robust Editing via Progressive Adaptive Intervention and Reintegration

## Abstract

Post-training for large language models (LLMs) is constrained by the high cost of acquiring new knowledge or correcting errors and by the unintended side effects that frequently arise from retraining. To address these issues, we introduce RE-PAIR (**R**obust **E**diting via **P**rogressive **A**daptive **I**ntervention and **R**eintegration), a lifelong editing framework designed to support precise and low-cost model updates while preserving non-target knowledge. REPAIR mitigates the instability and conflicts of large-scale sequential edits through a closed-loop feedback mechanism coupled with dynamic memory management. Furthermore, by incorporating frequent knowledge fusion and enforcing strong locality guards, RE-PAIR effectively addresses the shortcomings of traditional distribution-agnostic approaches that often overlook unintended ripple effects. Our experiments demonstrate that REPAIR boosts editing accuracy by 10%-30% across multiple model families and significantly reduces knowledge forgetting. This work introduces a robust framework for developing reliable, scalable, and continually evolving LLMs.

## 1 Introduction

Large language models (LLMs) have demonstrated remarkable capabilities across diverse tasks. However, their inherent rigidity prevents them from autonomously updating knowledge after pre-training, rendering them unable to correct errors (*e.g.*, hallucinations or outdated facts) or integrate new information. Consequently, lifelong model editing has emerged as a critical research paradigm. It aims to enable continuous, efficient, and low-cost local updates that ensure models remain accurate and relevant over time Wang et al. (2024b). In contrast to full re-training or broad fine-tuning, editing focuses on *precisely fine-grained* modifications that preserve unrelated competencies while delivering immediate corrections at deployment time.

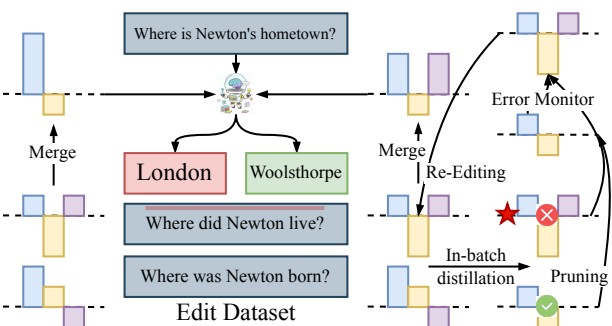

Figure 1: **Problems and our solutions**. REPAIR achieves closed-loop feedback, fine-grained knowledge integration, weighted knowledge merging, and robust editing performance.

Despite steady progress, important gaps remain as shown in Figure 1. **(1) Large-scale sequential editing & coarse knowledge fusion**. As edits accumulate, models can exhibit routing instability, conflicts among edits, and even collapse. Thus, stabilizing sequential updates without broad side effects remains challenging Gupta et al. (2024); Cohen et al. (2024). Semi-parametric designs (*e.g.*, SERAC Mitchell et al. (2022b)) and discrete key–value adaptors (*e.g.*, GRACE Hartvigsen et al. (2023)) alleviate some failure modes and support long edit streams, but still face scope and auditing trade-offs Mitchell et al. (2022b); Hartvigsen et al. (2023). The strategy for knowledge fusion remains underexplored, despite being the stage most prone to information loss Wang et al. (2024a). **(2) Few-shot editing**. Under data-scarce conditions, editors often struggle to form robust, generalizable

changes beyond the exact prompt, motivating gradient-transformation editors trained for locality (*e.g.*, MEND Mitchell et al. (2022a)) and broader taxonomies of edit generalization Mitchell et al. (2022a); Wang et al. (2024b). **(3) Open-loop and distribution-agnostic learning**. Many pipelines operate without reflective feedback, optimize on indiscriminate batches, and under-stress-test ripple effects on related knowledge and reasoning, calling for tighter evaluation and integration mechanisms Cohen et al. (2024); Wang et al. (2024b). Overall, these issues reveal a fundamental trade-off among reliability, specificity, and scalability that any practical editing system must reconcile.

To address these challenges, we propose the framework named REPAIR (**R**obust **E**diting via **P**rogressive **A**daptive **I**ntervention and **R**eintegration), with targeted strategies: **(1) Closed-loop feedback with dynamic memory management** that monitors edit performance and selectively re-initializes underperforming modules to stabilize routing and consolidation at scale. Concretely, our controller triggers health checks after each edit window and performs scoped resets or compaction when drift is detected. **(2) Distribution-aware optimization** that reorganizes samples by similarity and applies inner-batch distillation to enhance consistency and robustness in few-shot settings, encouraging edits to generalize across paraphrases and nearby contexts rather than overfitting to single prompts. **(3) Frequent knowledge fusion** that increases fusion cadence to prevent information loss and ensure timely consolidation of new and existing knowledge, with guardrails that validate locality before integration to avoid unintended side effects.

We compare REPAIR with several foundational model editing methods across three dimensions: *Memory*, *Attributes*, and *Behaviors* (Table 1). Its core innovation lies in integrating a dual memory system with parametric editing, complemented by error feedback, inner-batch knowledge distillation, and loss-aware subspaces merging. This design achieves high success rates and broad editing coverage while minimizing side effects. In contrast, previous methods struggle with knowledge overlap and loss, particularly in sequential editing, where large differences between adjacent samples hinder effective correction. Table 2 showcases cases where REPAIR outperforms baselines, offering a better balance of Reliability, Generalization, and Locality.

Table 1: **Comparison of current model editing methods.** "✓" refers to "yes" and "well-supported", "×" refers to "no" or "badly-supported", and "◯" refers to "less-supported". The three metrics of Reliability, Generalization, and Locality denote the performance on lifelong editing.

| Methods | Memory | | | Attributes | | | | Behaviors | |
|---|---|---|---|---|---|---|---|---|---|
| | Long-term Memory | Working Memory | Parametric | Lifelong | Reliability | Generalization | Locality | Error Feedback | Knowledge Distillation |
| FT-EWC Kirkpatrick et al. (2017) | ✓ | × | ✓ | ✓ | ✓ | ✓ | × | × | × |
| ROME Meng et al. (2022b) | ✓ | × | ✓ | × | × | × | × | × | × |
| MEMIT Meng et al. (2023) | ✓ | × | ✓ | × | × | × | × | × | × |
| MEND Mitchell et al. (2022a) | ✓ | × | ✓ | × | × | × | × | × | × |
| DEFER Mitchell et al. (2022b) | × | ✓ | ✓ | ✓ | ◯ | × | × | × | × |
| GRACE Hartvigsen et al. (2023) | × | ✓ | × | ✓ | ✓ | × | ✓ | × | × |
| WISE Wang et al. (2024a) | ✓ | ✓ | ✓ | ✓ | ✓ | ✓ | ✓ | × | × |
| REPAIR | ✓ | ✓ | ✓ | ✓ | ✓ | ✓ | ✓ | ✓ | ✓ |

Table 2: **Failure cases study**.Previous baselines((Wang et al., 2024a), (Hartvigsen et al., 2023))often encounter issues of repeating answers from previous questions and difficulty in correcting adjacent knowledge during editing.

| MethodPrompt | Edit Target | Post-Edit Output | Metrics |
|---|---|---|---|
| a) The genus Platypatrobus is part of the family? | Arctiinae | Arctiuc ✗ | Reliability✗ |
| b) *The genus Platypatrobus is a part of what family* | - | Yemen ✗ | Generalization✗ |
| c) The genus Platypatrobus is part of the family? | - | Arctiinae ✓ | |
| c) *When was the IAAF Combined Events Challenge launched?* | 2006 | Armand ✗ | Reliability✗ |
| d) *When does season 5 of ruby come out?* | October 14, 2017 | 2006 ✗ | Locality✗ |
| e)*when does season 5 of ruby come out?* | - | 2017✓ | |

In summary, the main contributions are as follows.

- We identify three critical challenges in model editing: (1) instability under large-scale sequential edits, (2) limited generalization in few-shot scenarios, and (3) inefficiency in open-loop, distribution-agnostic pipelines.

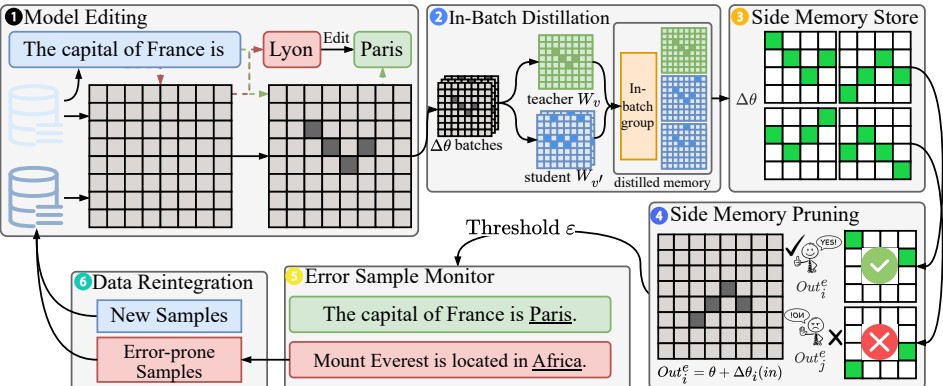

Figure 2: **The overall structure of REPAIR.** An edit, such as changing the capital of France from "Lyon" to "Paris," is stored as a parameter update, $\Delta\theta$, in the Side Memory. An Error Sample Monitor evaluates the performance of each edit ($Out_i^e$). If the error rate, $Err_{thresh}$, for an edit on a new sample exceeds a threshold $\epsilon$, the Side Memory Pruning module removes the erroneous update. The system then reintegrates new and error-prone samples for continuous learning.

- We propose REPAIR, a novel framework to address these challenges by integrating a dual-memory system with parametric editing. It introduces closed-loop error feedback, distribution-aware optimization, and loss-aware subspaces merging to ensure robust and precise updates.

- We validate the performance of REPAIR across diverse models (including LLaMA-3, Qwen-2.5, DeepSeek-R1-1.5B, and GPT-2-XL), demonstrating a 15%–20% improvement in overall editing performance over state-of-the-art methods and showing consistent, robust generalization.

## 2 METHODOLOGY

We propose a novel closed-loop lifelong model editing framework, denoted as **REPAIR**, which addresses the limitations of open-loop editing in distributed side-memory methods. Our framework, as shown in Figure 2, integrates (1) closed-loop error feedback with dynamic memory management; (2) distribution-aware batch reassembly with inner-batch knowledge distillation; (3) loss-aware weighted knowledge merging.

### 2.1 PROBLEM SETUP

**Definition 2.1** (Lifelong Model Editing). *Given a pre-trained model $f_{\theta_0}(y|x)$, a sequential edit stream $\{\mathcal{E}_t\}_{t=1}^{T}$ where $\mathcal{E}_t = \{(x_i^{(t)}, y_i^{(t)})\}_{i=1}^{N}$, and auxiliary distributions $\mathcal{G}(x)$ (paraphrased inputs) and $\mathcal{U}$ (unrelated contexts), the objective is to obtain updated parameters $\theta_T$ that optimize the multi-objective trade-off:*

$$\theta_t = \arg\min_{\theta} \alpha \underbrace{\frac{1}{N}\sum_{i=1}^{N} \ell\left(f_\theta(\cdot|x_i^{(t)}), y_i^{(t)}\right)}_{reliability} + \beta \underbrace{\frac{1}{N}\sum_{i=1}^{N} \mathbb{E}_{x'\sim\mathcal{G}(x_i^{(t)})}\left[\ell\left(f_\theta(\cdot|x'), y_i^{(t)}\right)\right]}_{generalization}$$

$$+ \gamma \underbrace{\mathbb{E}_{x\sim\mathcal{U}}\left[\mathrm{KL}\left(f_{\theta_{t-1}}(\cdot|x) \parallel f_\theta(\cdot|x)\right)\right]}_{locality} + \underbrace{R(\theta, \theta_{t-1})}_{stability} \tag{1}$$

*where $(\alpha, \beta, \gamma)$ are hyperparameters controlling the reliability-generalization-locality-stability trade-off, and $R$ denotes a regularization term enforcing parameter smoothness across sequential edits.*

## 2.2 DUAL MEMORY MECHANISM AND ROUTING

As shown in Figure 2, block 1: For dual memory-based editing methods, the dual memory mechanism is typically deployed in the deep layers of the network. Specifically, for the value matrix $\mathbf{W}_v$ of the target FFN layer, here create a copy as the side memory pool $M_s$, i.e.: $M_s^{(0)} = W_v$ If the side memory pool is activated, the output is computed as: $o_s = \phi(f^T W_k) \cdot M_s$, where $\phi$ denotes the non-linear activation function, and $o_s$ represents the FFN output based on the side memory(Wang et al. (2024a)).

During the inference phase, for memory pool $i$, the activation score is defined as

$$\Delta_{\text{act}}^{(i)}(x) = \|\mathcal{A}(x) \cdot (W_{v,i}' - W_v)\|_2. \tag{2}$$

where $\mathcal{A}(\cdot) = a$ is the activation of the side memory's corresponding FFN layer. Routing selects the pool with the activation score. If $\max_i \Delta_{\text{act}}^{(i)}(x) \leq \varepsilon$, the main memory $W_v$ is used. Otherwise, the side memory pool $M_s$ is selected. To enforce discriminative routing, we use a margin-based loss. The objective of the routing mechanism is to establish a clear decision boundary:

$$\min_{\mathbf{x}_e \sim \mathcal{E}} \mathcal{R}(\mathbf{x}_e) \sim \min_{\mathbf{x}' \sim \mathcal{U}} \mathcal{R}(\mathbf{x}') > \tau > \max_{\mathbf{x}_i \sim \mathcal{G}} \mathcal{R}(\mathbf{x}_i) \tag{3}$$

where $\tau$ is a preset threshold, and $\mathcal{E}$ and $\mathcal{G}_i$ represent the edit and edit-irrelevant datasets, respectively. This selective activation mechanism ensures that edited knowledge is only retrieved in relevant contexts, thereby minimizing interference with the original model's performance.

## 2.3 DISTRIBUTION-AWARE INNER-BATCH KNOWLEDGE DISTILLATION

As shown in Figure 2 block 2: A sample batch $\mathcal{E} = \{x_1, x_2, \ldots, x_n\}$, and denote the corresponding feature representations by $o_i = \text{Norm}(f_\theta(x_i), \quad i = 1, \ldots, n,)$. To improve the consistency and stability of model updates during sequential edits, we organized samples into homogeneous batches and performed intrabatch knowledge distillation. Samples with high mutual similarity are grouped into a batch $B = \{x^{(0)}, x^{(1)}, \ldots, x^{(b-1)}\}$. Within each batch, the first sample $x^{(0)}$ acts as a *teacher*, while the remaining samples are *students*. We define the inner-batch knowledge distillation loss as

$$\mathcal{L}_{\text{kd}} = \lambda \cdot \mathcal{L}_{\text{cosine}} + \theta \cdot \mathcal{L}_{\text{variance}} \tag{4}$$

where $\mathcal{L}_{\text{cosine}} = 1 - \frac{o_i \cdot o_0}{\|o_i\|\|o_0\|}$ and $\mathcal{L}_{\text{variance}} = \frac{1}{N} \sum_{i=1}^{N} \|o_i - o_{\text{mean}}\|^2$. Minimizing $\mathcal{L}_{\text{kd}}$ encourages all samples in the batch to share similar knowledge, which in turn reduces potential conflicts when updating the same network parameters $\theta$. The regularization term is used to maintain diversity among features, preventing excessive uniformity.

If certain samples cannot be well-aligned with the batch (i.e., their $\mathcal{L}_{\text{kd}}$ remains high after optimization), this indicates that they do not belong to the same distribution cluster and are unlikely to be effectively edited together. Such samples are removed from the batch and reclustered with other samples to form new homogeneous groups. Formally, the final batch reassembly can be expressed as

$$\mathcal{B}^* = \text{Recluster}\big(\{x \in B \mid \mathcal{L}_{\text{kd}}(x, B) < \epsilon\}\big), \tag{5}$$

where $\epsilon$ is a threshold controlling inner-batch consistency. This procedure ensures that sequential parameter edits are performed on groups of samples with aligned knowledge, improving both stability and effectiveness of the model update. The convergence proof is provided in the Appendix 4 and Appendix 2.

## 2.4 CLOSED-LOOP ERROR FEEDBACK AND MEMORY PRUNING

As shown in Figure 2 block 4: After each editing cycle, we evaluate the performance in a feedback pool $\mathcal{E}$ of error response samples by comparing to the correctness threshold $\tau_{\text{correct}}$. For each shard $i$, we define the error set $\mathcal{E}_i = \{x \in \mathcal{E} \mid i^*(x) = i\}$ and compute the error rate $r_i^{\text{pool}}$ for each side memory pool, defined as the proportion of failed edits within the corresponding sample set: $r_i^{\text{pool}} = \frac{|\{x \in \mathcal{E}_i \mid a(x) \leq \tau_{\text{correct}}\}|}{|\mathcal{E}_i|}$

When the pruning conditions are met ($r_i > \tau_{\text{prune}}$ or $|\mathcal{E}| > \tau_E$), we execute the following procedure:

1. **Memory pool screening & pruning:** Identify the side memory pool with the highest error rate $j = \arg\max_i r_i^{\text{pool}}$. Remove the identified memory pool from the system.

2. **Sample Reintegration & retraining:** Recombine the remaining error samples to form a new training set $\mathcal{E}_{\text{retrain}}$. Retrain the new side memory pools using $\mathcal{E}_{\text{retrain}}$.

This closed-loop feedback mechanism enables the system to dynamically identify and eliminate underperforming memory units while optimizing the overall editing performance through sample reorganization and iterative retraining. The time-convergence proof is provided in the Appendix 2.

### 2.5 MERGING WITH WEIGHTED TIES

As shown in Figure 2 block 3: After multiple updates, shards $\{W'_{v,i}\}$ produce deltas $\tau_i = W'_{v,i} - W_v$. We merge them with the weighted TIES Yadav et al. (2023) operator based on : $W'_v \leftarrow W_v + \omega_i \text{TIES}\left(\{\tau_i\}_{i=1}^k; W_v\right)$.

The total loss integrates all components:

$$\mathcal{L}_{\text{total}} = \mathcal{L}_{\text{edit}} + \lambda_a \mathcal{L}_a + \lambda_{\text{KD}} \mathcal{L}_{\text{KD}}. \tag{6}$$

$\mathcal{L}_{\text{edit}}$ is the autoregressive cross-entropy. $\mathcal{L}_{\text{edit}}(W'_v) = -\log P_{W'_v}(y \mid x)$. To enforce discriminative routing, we use a margin-based loss:

$$\mathcal{L}_a = \min \left\{ \max(0, \Delta_{\text{act}}(x_i) - \gamma_1) \right.$$
$$\left. + \max(0, \gamma_2 - \Delta_{\text{act}}(x_e)) + \max(0, \gamma - (\Delta_{\text{act}}(x_e) - \Delta_{\text{act}}(x_i))) \right\} \tag{7}$$

For shard $i$, consider $\|$ subspaces $\{\theta_1, \ldots, \theta_k\}$, each trained on a subset of samples $\mathcal{E}_i$. Let the average training loss of subspaces $\theta_i$ be: $\mathcal{L}_i = \frac{1}{|\mathcal{E}_i|} \sum_{(x,y)\in\mathcal{E}_i} \ell(f(x; \theta_i), y)$, where $\ell(\cdot)$ is the task loss. We define the merging weight of each subspaces as $w_i = \frac{\exp(-\alpha\mathcal{L}_i)}{\sum_{j=1}^M \exp(-\alpha\mathcal{L}_j)}$, with $\alpha > 0$ controlling sensitivity to the loss. The global network parameters are then obtained via weighted averaging: $\theta = \sum_{i=1}^M w_i \theta_i$. This loss-aware merging favors subspaces that achieve lower training loss on their corresponding samples, promoting reliable knowledge integration.

## 3 EXPERIMENTS

In the experimental section, we design six evaluations to answer the following questions: **Q1**, do the three key innovations (closed-loop feedback, discriminative pruning, and distribution reintegration) improve edit accuracy, generalization, and locality? **Q2**, does the method generalize well to knowledge-intensive tasks such as question answering and hallucination mitigation? **Q3**, is the method effective across different parameter scales and diverse architectures, including recent open-source models? **Q4**, under distribution shift (e.g., on the Wikibig Edit dataset), does the method remain robust and outperform existing methods? **Q5**, can the method maintain long-term stability and reliability in large-scale sequential editing scenarios? **Q6**, what are the contributions and sensitivities of each component and hyperparameter to overall performance?

### 3.1 EXPERIMENTAL SETUP

**Datasets and Models**. Autoregressive LLMs are ideal for evaluating model editing due to their unidirectional causal structure, which allows predictable and traceable edits. This ensures clear interpretability of edit generalization and locality. We evaluate widely used models (LLaMA-3-8B, GPT2-XL) and recent models (Qwen2.5-7B, DeepSeek-R1-1.5B), using datasets such as ZsRE for closed-book QA, Wikibig Edit for editing performance, and a hallucination dataset to assess generalization. For more details, refer to the Appendix 5.

**Baselines.**

- **Direct Parameter Editors**: Directly modify model weights (e.g., **ROME** Gupta et al. (2024), **MEMIT** Meng et al. (2023), **MEMIT-mass** Meng et al. (2023)).

- **Hypernetwork-Based Editors**: Use an auxiliary network to generate parameter updates at inference (e.g., **MEND** Mitchell et al. (2022a)).

- **External Memory-Based Editors**: Leave the model unchanged and store edits in external memory, retrieved via a routing mechanism (e.g., **SERAC** Mitchell et al. (2022b), **GRACE** Hartvigsen et al. (2023), **WISE** Wang et al. (2024b)).

**Implementation Details.** experiments were conducted simultaneously using two GPUs: an A100 PCIe 80GB and an A100 SXM4 40GB. The code was implemented based on PyTorch 2.1, with modifications built upon the original EasyEditor framework. The specific hyperparameter settings are detailed in Appendix C.

**Evaluation Metrics** Each edited corpus instance comprises three components: the descriptor $k_e$ used to perform the edit, an irrelevant prompt-answer pair $k'_e$ to verify locality and a rephrase prompt $k_{loc}$ to evaluate generalization performance across different expressions. To comprehensively evaluate the optimization capability of the proposed method in addressing the continual learning trilemma, we employ four metrics—edit accuracy: $\textbf{Rel} = \frac{1}{N}\sum_{n=1}^{N}\mathrm{l}(f_{\omega_N}(\mathbf{x}_e^n) = \mathbf{y}_e^n)$, rephrase accuracy : $\textbf{Gen} = \frac{1}{N}\sum_{n=1}^{N}\mathrm{l}(f_{\omega_N}(\mathbf{x'}_e^n) = \mathbf{y}_e^n)$, locality : $\textbf{Loc} = \frac{1}{N}\sum_{n=1}^{N}\mathrm{l}(f_{\omega_N}(\mathbf{x}_{loc}^n) = f_{\omega_0}(\mathbf{x}_{loc}^n))$. We use the geometric mean of Rel., Gen., and Loc. to evaluate the overall editing performance, which balances metric sensitivity and interpretability, exhibits sensitivity to weak performance areas, and is suitable for scenarios where all three metrics are equally important. $\textbf{OP} = \sqrt[3]{\text{Rel.} \times \text{Gen.} \times \text{Loc.}}$ to assess the holistic editing effectiveness. Here, $\mathrm{l}(\cdot)$ is the indicator function used to count the number of successful predictions.

For the hallucination dataset specifically, we utilize perplexity(PPL) as the metric to assess editing performance. PPL can be interpreted as the "average branching factor in predicting the next token," where a lower value indicates more accurate model predictions and suggests a reduced likelihood of the edited model generating hallucinations. $\textbf{PPL} = \exp\left(-\frac{1}{N}\sum_{i=1}^{N}\log P(y_i|\text{context}_i)\right)$

## 3.2 MAIN RESULTS

Table 3 effectively addressed **Q1**, **Q4** and **Q5**. It has been rigorously evaluated across diverse models and scales (N = 1, 30, 120, 1000) of QA editing tasks, demonstrating state-of-the-art performance. Fine-tuning-based methods achieve good accuracy and generalization at small scales but suffer from catastrophic forgetting and knowledge conflicts in large-scale edits, leading to performance degradation. GRACE excels in accuracy but has limited generalization, while WISE maintains strong locality but sacrifices critical knowledge, reducing editing accuracy. ROME-style methods are stable but overfit and struggle with generalization.

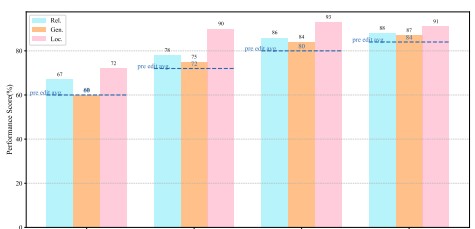

Figure 3: Average Editing Performance of Wik-iBigEdit Across Different Models

To address **Q2**, Table 4 shows REPAIR's effectiveness in reducing hallucinations on the SelfCheck-GPT dataset for LLaMA-3-8B across different editing scales. REPAIR balances reduced hallucinations with preserved locality, making it highly effective for large-scale model editing.

To address **Q3** and **Q4**, Table 3 and Figure 3 show that REPAIR's closed-loop error feedback, together with distribution-aware clustering and redistribution, yields consistently superior performance across edit scales and exceptional stability for large-scale edits. Smaller models concentrate knowledge in narrower parameter subsets, enabling reliable local corrections but weakening long-term stability and generalization (i.e., maintaining accuracy while preserving unrelated knowledge). Accordingly, DeepSeek-R1-1.5B attains higher immediate correction rates at small edit_Num, yet degrades quickly as N grows. For locality, LLaMA-3-8B and Qwen2.5-7B are marginally stronger due to parameter redundancy; DeepSeek-R1-1.5B remains competitive only at low N, then collapses under extreme multi-point editing. In contrast, larger models distribute knowledge more broadly, and though harder to modify—successful edits generalize better across contexts. At a medium scale (N=120), MEMIT-M and WISE show higher Rel., likely because REPAIR's

Table 3: Comparative results for QA on multi-scale editing (ZsRE and WikiBigEdit) $N$: Num Edits.

| Method | $N = 1$ | | | | $N = 30$ | | | | $N = 120$ | | | | $N = 1000$ | | | |
|---|---|---|---|---|---|---|---|---|---|---|---|---|---|---|---|---|
| | Rel. | Gen. | Loc. | OP. | Rel. | Gen. | Loc. | OP. | Rel. | Gen. | Loc. | OP. | Rel. | Gen. | Loc. | OP. |
| | | | | | | | LLaMA-3-8B (ZsRE) | | | | | | | | | |
| FT-L | 0.57 | 0.52 | 0.96 | 0.66 | 0.35 | 0.35 | 0.52 | 0.39 | 0.29 | 0.26 | 0.21 | 0.25 | 0.19 | 0.15 | 0.02 | 0.08 |
| FT-EWC | 0.96 | **0.93** | 0.02 | 0.26 | 0.78 | 0.76 | 0.02 | 0.23 | 0.76 | **0.76** | 0.08 | 0.36 | 0.69 | **0.67** | 0.08 | 0.33 |
| MEND | 0.95 | 0.93 | 0.96 | 0.95 | 0.24 | 0.25 | 0.18 | 0.22 | 0.08 | 0.07 | 0.00 | 0.00 | 0.00 | 0.00 | 0.00 | 0.00 |
| ROME | 0.85 | 0.80 | 0.99 | 0.88 | 0.61 | 0.60 | 0.68 | 0.63 | 0.22 | 0.22 | 0.04 | 0.12 | 0.01 | 0.01 | 0.01 | 0.01 |
| MEMIT-M | 0.84 | 0.81 | 0.99 | 0.88 | 0.73 | 0.72 | 0.95 | 0.79 | 0.70 | 0.65 | 0.82 | 0.72 | 0.63 | 0.63 | 0.62 | 0.63 |
| DEFER | 0.68 | 0.58 | 0.56 | 0.61 | 0.65 | 0.47 | 0.36 | 0.49 | 0.20 | 0.12 | 0.27 | 0.20 | 0.03 | 0.03 | 0.74 | 0.27 |
| GRACE | **0.97** | 0.36 | **1.00** | 0.71 | **0.96** | 0.17 | **1.00** | 0.55 | **0.94** | 0.14 | **1.00** | 0.51 | **0.93** | 0.08 | **1.00** | 0.42 |
| WISE | 0.94 | 0.92 | **1.00** | **0.95** | 0.62 | 0.60 | 0.86 | 0.68 | 0.57 | 0.58 | 0.87 | 0.66 | 0.45 | 0.44 | 0.51 | 0.47 |
| **REPAIR** | 0.94 | 0.92 | **1.00** | **0.95** | **0.93** | **0.90** | 0.87 | **0.89**↑ | 0.76 | 0.74 | **1.00** | **0.83**↑ | 0.68 | 0.65 | 0.89 | **0.73**↑ |
| | | | | | | | Qwen2.5-7B (ZsRE) | | | | | | | | | |
| FT-L | 0.68 | 0.63 | 0.93 | 0.74 | 0.28 | 0.23 | 0.44 | 0.30 | 0.13 | 0.11 | 0.10 | 0.11 | 0.08 | 0.06 | 0.02 | 0.05 |
| FT-EWC | 0.97 | 0.92 | 0.05 | 0.35 | 0.82 | 0.80 | 0.02 | 0.24 | 0.71 | 0.69 | 0.05 | 0.29 | 0.58 | 0.56 | 0.03 | 0.21 |
| MEND | 0.96 | **0.95** | 0.96 | 0.96 | 0.31 | 0.31 | 0.27 | 0.29 | 0.15 | 0.14 | 0.03 | 0.09 | 0.02 | 0.02 | 0.00 | 0.00 |
| ROME | 0.90 | 0.89 | 0.99 | 0.93 | 0.77 | 0.73 | 0.52 | 0.66 | 0.31 | 0.28 | 0.03 | 0.14 | 0.01 | 0.02 | 0.00 | 0.00 |
| MEMIT-M | 0.84 | 0.81 | 0.99 | 0.88 | 0.73 | 0.72 | 0.95 | 0.79 | 0.70 | 0.65 | 0.82 | 0.72 | 0.52 | 0.51 | 0.57 | 0.53 |
| DEFER | 0.74 | 0.67 | 0.88 | 0.76 | 0.58 | 0.51 | 0.44 | 0.51 | 0.22 | 0.21 | 0.43 | 0.27 | 0.14 | 0.08 | 0.25 | 0.14 |
| GRACE | 0.97 | 0.41 | 0.98 | 0.73 | **0.97** | 0.2 | **1.00** | 0.58 | **0.95** | 0.08 | **0.98** | 0.42 | **0.94** | 0.02 | **1.00** | 0.27 |
| WISE | 0.97 | **0.95** | 0.98 | 0.97 | 0.79 | 0.73 | 0.91 | 0.80 | 0.59 | 0.57 | 0.92 | 0.68 | 0.44 | 0.41 | 0.72 | 0.51 |
| **REPAIR** | **0.98** | **0.95** | **1.00** | **0.98** ↑ | 0.93 | **0.90** | 0.93 | **0.92**↑ | 0.81 | **0.80** | 0.92 | **0.84**↑ | 0.72 | **0.70** | 0.67 | **0.69**↑ |
| | | | | | | | DeepSeek-R1-1.5B (WikiBigEdit) | | | | | | | | | |
| FT-L | 0.71 | 0.68 | 0.93 | 0.77 | 0.26 | 0.20 | 0.76 | 0.34 | 0.13 | 0.11 | 0.37 | 0.17 | 0.02 | 0.02 | 0.08 | 0.03 |
| FT-EWC | 0.93 | 0.91 | 0.33 | 0.65 | 0.70 | 0.70 | 0.18 | 0.45 | 0.42 | 0.41 | 0.07 | 0.23 | 0.18 | 0.15 | 0.02 | 0.08 |
| MEND | 0.91 | 0.87 | 0.95 | 0.91 | 0.43 | 0.38 | 0.10 | 0.25 | 0.24 | 0.23 | 0.08 | 0.16 | 0.03 | 0.03 | 0.02 | 0.05 |
| ROME | 0.86 | 0.83 | 0.97 | 0.88 | 0.72 | 0.71 | 0.67 | 0.70 | 0.18 | 0.18 | 0.02 | 0.09 | 0.01 | 0.0 | 0.01 | 0.00 |
| MEMIT-M | 0.86 | 0.87 | 0.97 | 0.90 | 0.78 | 0.77 | 0.82 | 0.79 | 0.54 | 0.51 | 0.77 | 0.60 | 0.38 | 0.38 | 0.62 | 0.45 |
| DEFER | 0.68 | 0.58 | 0.47 | 0.35 | 0.63 | 0.61 | 0.51 | 0.58 | 0.17 | 0.15 | 0.33 | 0.20 | 0.07 | 0.07 | 0.12 | 0.08 |
| GRACE | 0.96 | 0.47 | **0.99** | 0.76 | **0.93** | 0.24 | **0.91** | 0.59 | **0.76** | 0.13 | 0.89 | 0.44 | 0.63 | 0.07 | **0.81** | 0.33 |
| WISE | 0.89 | 0.91 | 0.98 | 0.93 | 0.76 | 0.74 | 0.89 | 0.79 | 0.64 | 0.65 | 0.83 | 0.70 | 0.47 | 0.38 | 0.61 | 0.48 |
| **REPAIR** | **0.98** | **0.93** | 0.98 | **0.96**↑ | 0.84 | **0.83** | 0.91 | **0.86**↑ | 0.71 | **0.69** | 0.90 | **0.76**↑ | 0.58 | 0.54 | 0.81 | **0.63**↑ |

Table 4: Main editing results for Hallucination task (SelfCheckGPT).

| Method | $N = 1$ | | $N = 30$ | | $N = 120$ | | $N = 500$ | |
|---|---|---|---|---|---|---|---|---|
| | Rel. (*PPL* ↓) | Loc. (↑) | Rel. (↓) | Loc. (↑) | Rel. (↓) | Loc. (↑) | Rel. (↓) | Loc. (↑) |
| | | | LLaMA-3-8B | | | | | |
| FT-L | 4.27 | 0.96 | 3.15 | 0.71 | 34.52 | 0.43 | 51.31 | 0.26 |
| FT-EWC | 2.18 | 0.24 | 3.51 | 0.09 | 2.90 | 0.21 | 3.48 | 0.24 |
| MEND | 5.34 | 0.87 | 1.24 | 0.86 | 9.17 | 0.89 | 564.9 | 0.00 |
| ROME | 1.88 | 0.99 | 2.47 | 0.94 | 84.56 | 0.03 | 73.4 | 0.02 |
| MEMIT-M | 1.62 | **1.00** | 1.78 | 0.99 | 8.03 | 0.99 | 7.43 | 0.94 |
| DEFER | **1.29** | 0.23 | 4.12 | 0.28 | 8.91 | 0.19 | 15.16 | 0.12 |
| GRACE | 2.21 | **1.00** | 8.67 | **1.00** | 7.24 | **1.00** | 6.18 | **1.00** |
| WISE | 1.91 | **1.00** | 1.59 | **1.00** | 1.14 | 0.99 | 2.08 | 0.99 |
| **REPAIR** | 1.43 | **1.00** | 1.37 | **1.00** | **1.12** | **1.00** | 1.91 | **1.00** |

pruning/reassembly introduces transient instability before sufficient error signals accumulate; however, at N=1000 their performance drops sharply, while REPAIR's dynamic adjustment preserves robustness and achieves the best overall metric. The error distribution can be seen in Appendix.

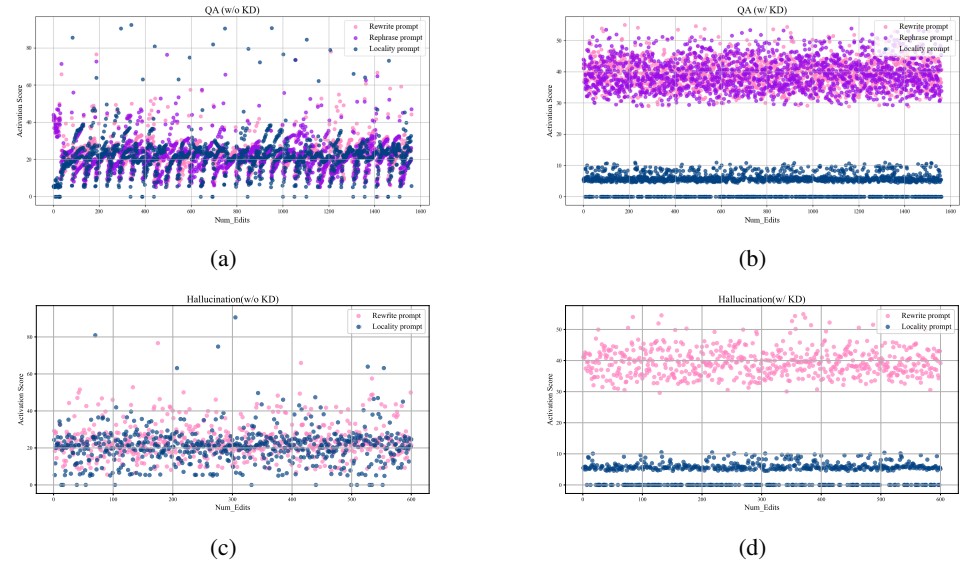

(a)

(b)

(c)

(d)

Figure 4: **Activation Score Visualization**. Results on LLaMA-3 for the WikiBigEdit dataset (N=1550) for the QA task and the SelfCheckGPT dataset for hallucination (N=600).

Figure 3 further addresses **Q1** regarding the effectiveness of distillation. For external memory-based editors, the ability to select the correct network for inference directly determines editing performance. The activation score, which serves as a critical routing criterion in memory networks, must exhibit statistically significant differences between new knowledge and irrelevant knowledge to ensure both reliability and locality of edits. As shown in Figure 4 (a) and (c), prior methods relying solely on triple-boundary loss fail to adequately separate the activation scores of $Data_{edit}$, $Data_{rephrase}$, and $Data_{loc}$, particularly in large-scale continual editing scenarios, leading to a breakdown of the routing mechanism. This deficiency fundamentally limits their editing performance. In contrast, by introducing inner-batch knowledge distillation, sample filtering, and samples reintegration, KD, as shown in Figure 4 (b) and (d), achieves a clear separation among the three types of samples, thereby ensuring the proper functioning of the routing mechanism.

## 3.3 ABLATION STUDIES

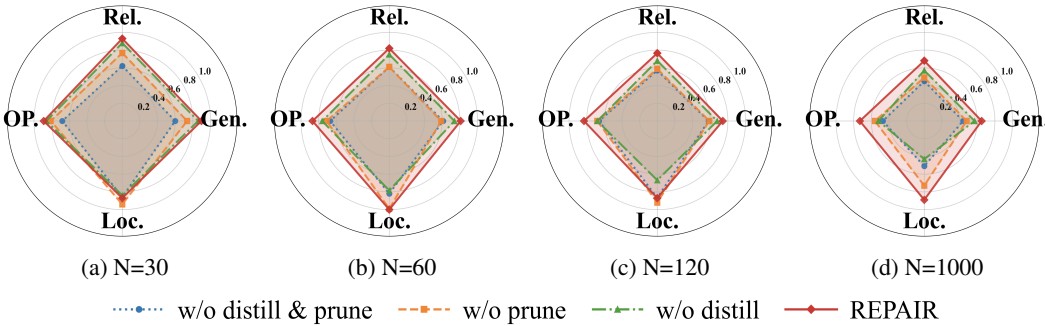

(a) N=30          (b) N=60          (c) N=120          (d) N=1000

········· w/o distill & prune    ─ ■ ─ w/o prune    ─ ▲ ─ w/o distill    ─ ◆ ─ REPAIR

Figure 5: **Performance comparison of different components.** Each radar chart shows performance on four metrics: Rel., gen., loc., and OP. on Qwen2.5 with ZsRE.

The evaluation of the overhead and throughput of REPAIR can be found in the Appendix 7. To answer **Q6**, we conducted comprehensive evaluations across four dimensions to assess the effectiveness of each component of REPAIR and analyze critical hyperparameter sensitivity under different editing scales. Notably, REPAIR demonstrates robustness in large-scale editing scenarios

that prior methods fail to achieve. As the number of edits increases, REPAIR exhibits increasingly pronounced advantages in overall performance: effective routing ensures strong locality, while the error-feedback mechanism maintains continual reliability. As shown in Figure 5 (a)–(d), the relative contributions of REPAIR's components vary across sample regimes but complement each other seamlessly. In small-scale edits, pruning with error feedback substantially improves reliability, while in large-scale scenarios, distribution-aware recognition and knowledge distillation become more critical. Regarding hyperparameter analysis in Figure 6, we observe distinct performance patterns: low thresholds fail to filter low-quality samples, limiting corrective opportunities; The total number of edits is limited, and the filtered erroneous samples cannot receive sufficient corrective training, which limits overall performance. A large number of erroneous samples in the early stage undergo continuous learning, causing the model to quickly fall into local optima, leading to catastrophic degradation of generalization. Subsequent learning yields minimal improvement, resulting in poor performance. In the upper-right quadrant, the absence of error feedback leaves many suboptimal samples, and the model editing efficiency is relatively high, approximating an open-loop editing process. In the lower-right quadrant, the model training efficiency is the lowest, but excessive editing can introduce overfitting risks, wasting computational resources on edits with low marginal utility.

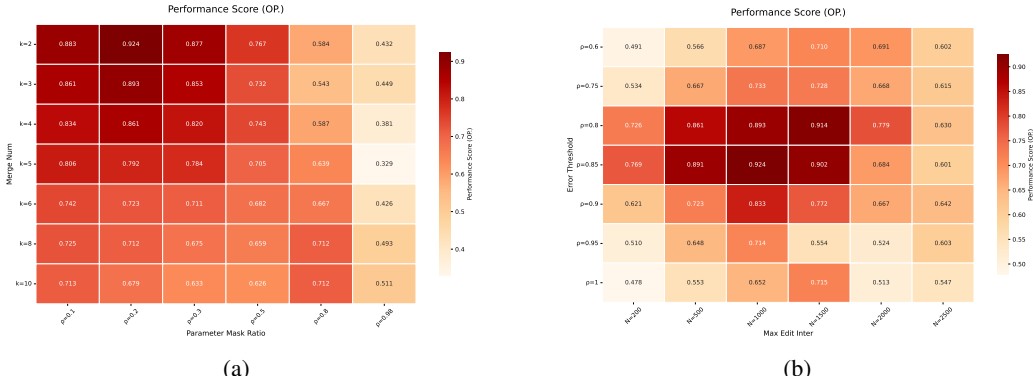

(a)              (b)

Figure 6: **Performance heatmap for the N=120 QA task on the LLaMA3 model.** Figure (a) shows the sensitivity analysis of two hyperparameters: the number of subspaces and the amount of updated parameters; Figure (b) analyzes the impact of error threshold and maximum iteration count on performance, with optimal performance observed at intermediate values.

## 4 CONCLUSION

In this work, we proposed **REPAIR**, a robust framework for lifelong model editing integrating error closed-loop feedback, inner-batch knowledge distillation, and loss-aware subspaces merging. Extensive experiments demonstrate that REPAIR maintains high performance under small-scale edits and exhibits remarkable robustness in large-scale editing scenarios, consistently outperforming existing baselines. These results highlight the potential of combining memory-aware strategies with optimization-driven editing for reliable and precise model updates. The intra-group distillation explicitly encourages feature alignment among similar samples, guiding the elimination and recombination of inconsistent samples. The loss-aware merging assigns higher weights to subspaces achieving lower training loss, effectively preserving reliable knowledge and reducing information dilution. Extensive experiments show that REPAIR consistently improves reliability and generalization, and demonstrates clear advantages in large-scale editing scenarios, highlighting the effectiveness of coordinated sample-level alignment and global reliability-aware merging.

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

# A STATEMENT

## A.1 ETHICS STATEMENT

This work studies safe, auditable editing of large language models using only publicly available datasets (ZsRE, WikiBigEdit, and a hallucination set) and off-the-shelf pretrained models; no human subjects or personally identifiable data were collected. We follow all dataset/model licenses and the double-blind review policy. Potential risks include misuse of editing to inject misinformation or to weaken safety constraints, and unintended spillover of edits to unrelated behaviors. To mitigate these risks, our framework emphasizes locality and closed-loop error checks before and after integration, and we report reliability–generalization–and locality metrics to surface side effects. Upon release, we will include guardrails such as edit logs, validation suites, reversible edits, and instructions for responsible use. These design choices align with REPAIR's stated goal of precise updates with locality safeguards.

## A.2 REPRODUCIBILITY STATEMENT

We will release our code, configs, and seeds to reproduce all results end-to-end after acceptance. Scripts fetch data/models, fix environments, and regenerate all tables/figures with the exact metrics (Rel./Gen./Loc./OP., PPL); hardware and hyperparameters are documented.

## A.3 AI USAGE STATEMENT

We used large language model–based tools during writing and implementation for text polishing, grammar and usage checks, and programming assistance (e.g., example code, refactoring, comments, and script templates). All AI-generated suggestions were reviewed, revised, and validated by the authors. The experimental design, data processing, result analysis, and conclusions were conducted independently by the authors; AI tools do not constitute authorship or academic credit. No sensitive or restricted data were provided to the tools, and they were not used to automatically generate experimental results or to replace essential human judgment.

# B  RELATED WORK

## B.1  CONTINUAL LEARNING

Continual Learning (CL)—also known as Incremental Learning or Lifelong Learning—aims to enable models to learn sequentially from a stream of tasks without forgetting previously acquired knowledge. The core challenge in CL is catastrophic forgetting, where adapting to new tasks leads to a significant degradation in performance on earlier tasks Kirkpatrick et al. (2017); McCloskey & Cohen (1989). To address this, numerous methods have been proposed, which can be broadly categorized into five groups: regularization-based, replay-based, optimization-based, representation-based, and architecture-based approaches.

Regularization-based methods mitigate forgetting by adding constraints to the loss function to preserve important parameters or behaviors from previous tasks. For example, Elastic Weight Consolidation (EWC) leverages Fisher information to regularize parameter updates Kirkpatrick et al. (2017), while Learning without Forgetting (LwF) uses knowledge distillation to maintain output consistency Li & Hoiem (2018).

Replay-based methods retain or generate samples from previous tasks to approximate old data distributions. Experience replay stores a subset of prior samples in a memory buffer Lopez-Paz & Ranzato (2017), whereas generative replay synthesizes pseudo-samples using deep generative models such as GANs or VAEs Shin et al. (2017).

Optimization-based methods manipulate the optimization process itself to avoid interference between tasks. Gradient Episodic Memory (GEM) projects gradients so as not to increase loss on previous tasks Lopez-Paz & Ranzato (2017), while Orthogonal Gradient Descent (OGD) promotes updates that are orthogonal to gradient directions associated with past tasks Farajtabar et al. (2020).

Representation-based methods focus on learning robust and transferable features that are less prone to forgetting. Self-supervised learning Fini et al. (2022) and large-scale pre-training Mehta et al. (2023) have been shown to bolster CL performance by providing more stable representations.

Architecture-based methods Ran et al. (2024); Rusu et al. (2016); Mallya & Lazebnik (2018)dynamically expand or partition the network to allocate task-specific parameters. Progressive Networks add new columns for each incoming task with lateral connections to prior columns Rusu et al. (2016), while PackNet iteratively prunes and reuses weights to free capacity for new tasks Mallya & Lazebnik (2018).

Recent trends extend CL to more realistic and challenging settings, including class-incremental learning (CIL), task-free CL (TFCL), online CL (OCL), and applications across object detection, semantic segmentation, reinforcement learning, and natural language processing Wang et al. (2023).

## B.2  MODEL EDITING

Model editing targets post-hoc modification of a trained model's behavior to insert, correct, or remove specific knowledge, ideally without harming unrelated capabilities. A common taxonomy distinguishes (i) *direct / training-free* parameter edits, (ii) *learning-based* editors that predict weight updates, and (iii) *semi-parametric* systems that externalize edits via retrieval or memory; recent surveys consolidate definitions, benchmarks, and open challenges Wang et al. (2024b).

ROME locates causal mediators of factual associations in mid-layer feed-forward (MLP) modules of Transformers and applies a rank-one update to edit a single fact Meng et al. (2022a). MEMIT extends this idea to *mass editing*, deriving multi-layer closed-form updates that scale to thousands of edits in large models while maintaining stronger locality than prior methods Meng et al. (2023). Although effective, subsequent analyses highlight stability issues under *sequential* edits and propose remedies Gupta et al. (2024).

Early work framed editing as learning a small hypernetwork to predict weight deltas from an edit specification: KnowledgeEditor (KE) learns constrained updates to change a model's factual prediction while preserving behavior on paraphrases Cao et al. (2021). MEND trains lightweight editor networks to transform fine-tuning gradients, enabling fast, local edits at scale across architectures

Table 5: Dataset statistics

| Task | Editing Data | N | Pre-edit(LLaMA/Qwen) | Locality Data |
|---|---|---|---|---|
| QA | ZsRE | 1000 | 0.25/0.21 ACC | NQ Kwiatkowski et al. (2019) |
| | WikiBigEdit | 500K | 0.36/0.32 ACC | NQ |
| Hallu. | SelfCheckGPT | 600 | 28.7/29.1 PPL | RedPajama Weber et al. (2024) |

Table 6: Hyperparameter settings

| HYPER | VALUE | HYPER | VALUE | HYPER | VALUE |
|---|---|---|---|---|---|
| ZsRE on LLaMA-3 | | | | | |
| Mask ratio | 0.20 | Edit_lr | 0.90 | Err_Thresh | 0.85 |
| $\lambda_a$ | 1.00 | $\lambda_{KD}$ | 1.00 | Max_iter | 10000 |
| Temperature | 2.00 | Act ratio | 0.20 | Layer_ID | 29.00 |
| $\gamma_1$ | 2.00 | $\gamma_2$ | 20.00 | $\gamma$ | 10.00 |
| $n_{\text{iter}}$ | 30.00 | $\lambda$ | 0.20 | Act_ratio | 0.30 |
| ZsRE on Qwen2.5 | | | | | |
| Mask ratio | 0.20 | Edit_lr | 0.90 | Err_Thresh | 0.85 |
| $\lambda_a$ | 2.00 | $\lambda_{KD}$ | 1.00 | Max_iter | 10000 |
| Temperature | 2.00 | Act ratio | 0.88 | Layer_ID | 23.00 |
| $\gamma_1$ | 5.00 | $\gamma_2$ | 20.00 | $\gamma$ | 10.0 |
| $n_{\text{iter}}$ | 50.00 | $\lambda$ | 0.30 | Act_ratio | 0.30 |
| Selfcheck GPT on LLaMA-3-8B | | | | | |
| Mask ratio | 0.20 | Edit_lr | 1.00 | Err_Thresh | 0.85 |
| $\lambda_a$ | 5.00 | $\lambda_{KD}$ | 1.00 | Max_iter | 5000 |
| Temperature | 2.00 | Act ratio | 0.88 | Layer_ID | 27.00 |
| $\gamma_1$ | 5.00 | $\gamma_2$ | 20.00 | $\gamma$ | 10.00 |
| $n_{\text{iter}}$ | 50.00 | $\lambda$ | 0.20 | Act_ratio | 0.80 |

Mitchell et al. (2022a). Instruction-driven variants further condition edits on natural-language instructions to improve usability and control Zhang et al. (2024).

Semi-parametric approaches such as SERAC store edits in an external key–value memory and learn to route between the base model and retrieved counterfactuals, achieving strong reliability and specificity without permanently altering base parameters Mitchell et al. (2022b). This design is attractive when edits must be audited, reverted, or scoped to contexts.

Editing methods are typically assessed along *reliability* (does the change take effect), *locality/specificity* (does unrelated behavior remain intact), and *generalization* (do edits transfer to paraphrases and contexts). Standard benchmarks include CounterFact and zsRE Meng et al. (2022c); Levy et al. (2017). Recent studies examine *ripple effects* beyond targeted facts, revealing broader side impacts on reasoning and distributed knowledge, and call for more rigorous, stress-testing evaluations Cohen et al. (2024). Overall, direct, learning-based, and semi-parametric approaches offer complementary trade-offs in edit scalability, controllability, and safety; combining precise localization with guardrails (e.g., retrieval gating, edit scopes, or validation filters) remains an active direction Wang et al. (2024b).

## C EXPERIMENTS DETAILS

The experiment details are given in Table 5, and hyperparameters are in Table 6.

Under identical hardware and batch configurations, the WISE baseline exhibits lower per-unit overhead. REPAIR demonstrates a similar scaling slope but with a higher intercept, primarily attributable to:

Table 7: Main results for QA on DeepSeek-R1-1.5B $N$: Num Edits.

| Method | $N = 1$ | | | | $N = 30$ | | | | $N = 120$ | | | | $N = 1000$ | | | |
|---|---|---|---|---|---|---|---|---|---|---|---|---|---|---|---|---|
| | Rel. | Gen. | Loc. | OP. | Rel. | Gen. | Loc. | OP. | Rel. | Gen. | Loc. | OP. | Rel. | Gen. | Loc. | OP. |
| DeepSeek-R1-1.5B (ZsRE) | | | | | | | | | | | | | | | | |
| FT-L | 0.43 | 0.42 | 0.95 | 0.56 | 0.32 | 0.33 | 0.46 | 0.36 | 0.21 | 0.21 | 0.15 | 0.19 | 0.17 | 0.15 | 0.09 | 0.13 |
| FT-EWC | 0.97 | **0.94** | 0.15 | 0.52 | 0.82 | 0.81 | 0.02 | 0.24 | 0.63 | 0.64 | 0.02 | 0.20 | 0.57 | 0.56 | 0.02 | 0.19 |
| MEND | 0.95 | **0.94** | 0.98 | 0.96 | 0.42 | 0.42 | 0.18 | 0.32 | 0.18 | 0.12 | 0.07 | 0.11 | 0.8 | 0.03 | 0.00 | 0.00 |
| ROME | 0.87 | 0.87 | 0.99 | 0.91 | 0.66 | 0.64 | 0.72 | 0.67 | 0.17 | 0.18 | 0.09 | 0.14 | 0.01 | 0.01 | 0.01 | 0.01 |
| MEMIT-M | 0.88 | 0.87 | 0.99 | 0.91 | 0.71 | 0.72 | 0.92 | 0.78 | 0.63 | 0.65 | 0.78 | 0.68 | 0.48 | 0.47 | 0.53 | 0.49 |
| DEFER | 0.62 | 0.60 | 0.82 | 0.67 | 0.58 | 0.57 | 0.57 | 0.57 | 0.34 | 0.31 | 0.23 | 0.29 | 0.07 | 0.06 | 0.02 | 0.04 |
| GRACE | **0.98** | 0.31 | 0.99 | 0.67 | **0.92** | 0.22 | **0.98** | 0.58 | **0.89** | 0.13 | **1.00** | 0.49 | **0.83** | 0.05 | **0.94** | 0.34 |
| WISE | 0.92 | 0.90 | **1.00** | 0.94 | 0.86 | 0.85 | 0.92 | 0.88 | 0.72 | 0.72 | 0.87 | **0.77** | 0.49 | 0.47 | 0.47 | 0.48 |
| **REPAIR** | 0.93 | 0.93 | **1.00** | **0.95** | 0.91 | **0.89** | 0.87 | **0.89**↑ | 0.74 | **0.74** | 0.82 | **0.77**↑ | 0.59 | **0.57** | 0.61 | **0.59**↑ |

Table 8: Main results for QA (ZeRE) on multi-model editing with error distribution.

| Method | $N = 1$ | | | $N = 30$ | | |
|---|---|---|---|---|---|---|
| | Rel. | Gen. | Loc. | Rel. | Gen. | Loc. |
| LLaMA-3-8B | $0.94 \pm 0.008$ | $0.92 \pm 0.01$ | $1.00^{+0.00}_{-0.02}$ | $0.93 \pm 0.003$ | $0.90 \pm 0.003$ | $0.87 \pm 0.004$ |
| Qwen2.5-7B | $0.98 \pm 0.02$ | $0.95 \pm 0.03$ | $1.00^{+0.00}_{-0.02}$ | $0.93 \pm 0.04$ | $0.90 \pm 0.03$ | $0.93 \pm 0.01$ |
| DeepSeek-R1 | $0.93 \pm 0.02$ | $0.92 \pm 0.03$ | $0.99 \pm 0.01$ | $0.91 \pm 0.01$ | $0.89 \pm 0.03$ | $0.87 \pm 0.01$ |
| GPT2-XL | $0.91 \pm 0.03$ | $0.92 \pm 0.03$ | $0.99 \pm 0.01$ | $0.88 \pm 0.03$ | $0.88 \pm 0.02$ | $0.84 \pm 0.01$ |

| Method | $N = 120$ | | | $N = 1000$ | | |
|---|---|---|---|---|---|---|
| | Rel. | Gen. | Loc. | Rel. | Gen. | Loc. |
| LLaMA-3-8B | $0.76 \pm 0.03$ | $0.74 \pm 0.02$ | $1.00^{+0.00}_{-0.04}$ | $0.68 \pm 0.05$ | $0.65 \pm 0.01$ | $0.89 \pm 0.04$ |
| Qwen2.5-7B | $0.81 \pm 0.04$ | $0.80 \pm 0.05$ | $0.92 \pm 0.03$ | $0.72 \pm 0.05$ | $0.70 \pm 0.04$ | $0.67 \pm 0.03$ |
| DeepSeek-R1 | $0.74 \pm 0.03$ | $0.74 \pm 0.04$ | $0.82 \pm 0.05$ | $0.59 \pm 0.02$ | $0.57 \pm 0.01$ | $0.61 \pm 0.03$ |
| GPT2-XL | $0.79 \pm 0.02$ | $0.77 \pm 0.01$ | $0.80 \pm 0.03$ | $0.61 \pm 0.03$ | $0.62 \pm 0.01$ | $0.68 \pm 0.02$ |

- Distribution-aware clustering and reorganization;

- Additional forward/backward passes for in-batch distillation;

- The triggering frequency and cost of closed-loop pruning and retraining at large scales;

- The final **merging (TIES)** cost.

As $N$ increases, the runtime curves of ROME, MEND, and FT-L exhibit significantly steeper growth, becoming substantially expensive or nearly infeasible at $N = 10^3$.

**Throughput** demonstrates that WISE maintains approximately $\sim 1.8$ edits/min at large $N$, followed by GRACE. REPAIR achieves $\sim 0.8$-$0.9$ edits/min at scale, lower than WISE and MEMIT-M, consistent with expectations given its additional computational procedures.

Error bars (representing standard deviation across multiple runs) indicate that REPAIR exhibits slightly higher variance than WISE, attributable to fluctuations in retriggering frequency and sample distribution characteristics.

**Relative overhead** shows the time ratio of REPAIR to WISE increasing from $\sim 1.6\times$ to $\sim 2.2\times$ with increasing scale.

## D  THEORETICAL ANALYSIS AND PROOF SKETCHES

We now provide theoretical justifications for the stability and convergence of the proposed REPAIR framework. We introduce formal assumptions and derive lemmas and theorems that characterize the behavior of our method.

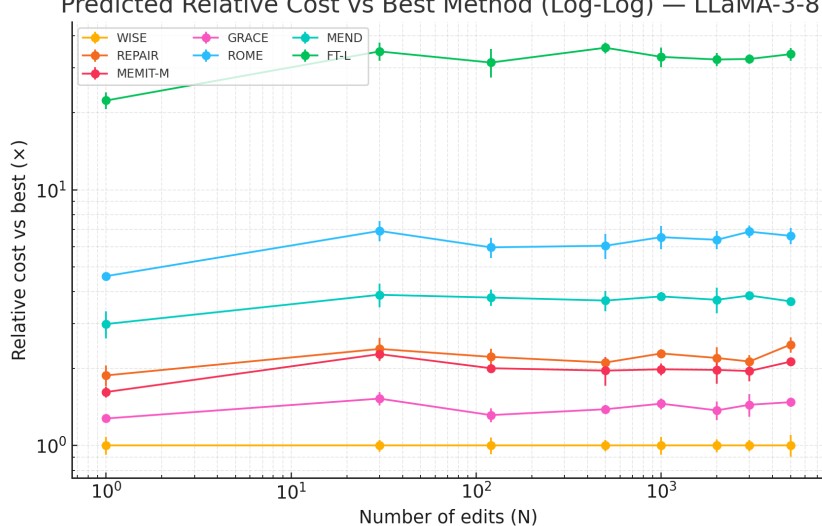

Figure 7: **Cost-Performance Assessment.** The total runtime of each method scales approximately linearly with the editing scale $N$, appearing as straight lines with slopes close to 1 in log-log coordinates. This indicates that the primary overhead is proportional to the number of edited entries.

### D.1 PRELIMINARIES

**Assumption 1** (Standard Optimization Setting). *We assume that the loss function $\mathcal{L}(\Theta)$ is $L$-smooth, i.e.,*

$$\|\nabla\mathcal{L}(\Theta_1) - \nabla\mathcal{L}(\Theta_2)\| \leq L\|\Theta_1 - \Theta_2\|,$$

*and bounded below by $\mathcal{L}^* > -\infty$. Learning rates satisfy $\eta_t > 0$ and $\sum_t \eta_t = \infty$, $\sum_t \eta_t^2 < \infty$.*

### D.2 STABILITY OF MASKED GRADIENT UPDATES

**Lemma 1** (Norm Bound under Masked Updates). *Let $g_i = \nabla_{W'_{v,i}}\mathcal{L}$ and $M_i$ be a Bernoulli mask. Then the masked update*

$$\Delta W'_{v,i} = -\eta(M_i \odot g_i)$$

*satisfies $\|\Delta W'_{v,i}\|_2 \leq \eta\|g_i\|_2$.*

*Proof.* Since $M_i$ is a coordinate projection, $M_i \odot g_i$ removes certain entries of $g_i$ and never increases its magnitude. Hence $\|M_i \odot g_i\|_2 \leq \|g_i\|_2$. Multiplying by $\eta$ yields the claim. □

**Theorem 1** (Inter-Shard Stability). *Assume masks $\{M_i\}$ are sampled independently with overlap probability $\rho^2$. Then in expectation,*

$$\mathbb{E}[\langle M_i \odot g_i, M_j \odot g_j\rangle] = \rho^2\langle g_i, g_j\rangle.$$

*Thus, masking reduces the expected conflict between gradients of different shards.*

*Proof.* For each coordinate $p$, $\Pr[M_i(p) = 1, M_j(p) = 1] = \rho^2$. Therefore, the expected inner product between masked gradients is $\rho^2$ times the original inner product. This reduces cross-shard interference and improves stability. □

### D.3 CLOSED-LOOP RE-TRIGGER ANALYSIS

**Assumption 2** (Error Reduction per Re-trigger). *Suppose that each re-trigger reduces the error rate of shard $i$ by at least a fixed constant $\delta > 0$, unless it is already below the pruning threshold $\tau_{\text{prune}}$.*

**Lemma 2** (Linear Error Decrease). *Let $r_i^{(n)}$ denote the error rate after $n$ re-triggers. Under Assumption 2,*

$$r_i^{(n)} \le r_i^{(0)} - n\delta.$$

**Theorem 2** (Finite-Time Convergence). *If $r_i^{(0)}$ is the initial error rate, then after at most*

$$N \ge \frac{r_i^{(0)} - \tau_{\mathrm{prune}}}{\delta}$$

*re-triggers, the error rate satisfies $r_i^{(N)} \le \tau_{\mathrm{prune}}$.*

*Proof.* By Lemma 3, $r_i^{(N)} \le r_i^{(0)} - N\delta$. Choosing $N$ such that $r_i^{(0)} - N\delta \le \tau_{\mathrm{prune}}$ ensures convergence below threshold in finite time. $\square$

### D.4 OVERALL CONVERGENCE INTUITION

**Theorem 3** (Closed-Loop Stability of REPAIR). *Under Assumptions 1 and 2, the iterative process combining masked updates, inner-batch distillation, and closed-loop re-trigger forms a contractive mapping in expectation. Consequently, the system converges to a stable edited state with a bounded error rate and without catastrophic forgetting.*

*Proof Sketch.* Masked updates reduce the variance of parameter updates, inner-batch distillation aligns outputs across samples, and re-trigger guarantees finite-time reduction of shard-level error rates. Together, these components yield monotone improvement. By standard stochastic contraction arguments, the process converges to a fixed point characterized by consistent batch predictions and an error rate below $\tau_{\mathrm{prune}}$. $\square$

**Lemma 3** (Zero-variance at any global minimizer). *Let $\mu = \frac{1}{m} \sum_{i=1}^{m} o_i$ and $\mathcal{L}_{\mathrm{var}} = \frac{1}{m} \sum_i \|o_i - \mu\|^2$. If not all $o_i$ are equal, then $\mathcal{L}_{\mathrm{var}} > 0$, while if $o_1 = \cdots = o_m = v$ (with $\|v\| = 1$) then $\mathcal{L}_{\mathrm{var}} = 0$. Hence every global minimizer of $\mathcal{L}_{\mathrm{KD}}$ on $(\mathbb{S}^{d-1})^m$ must satisfy $o_1 = \cdots = o_m =: v$.*

**Lemma 4** (Unique global minimizer). *Under the conclusion of Lemma 3, minimizing $\mathcal{L}_{\mathrm{KD}}(v) = \lambda(1 - \langle v, u \rangle)$ over $\|v\| = 1$ gives the unique solution $v^\star = u$. Therefore the unique global minimizer of $\mathcal{L}_{\mathrm{KD}}$ on $(\mathbb{S}^{d-1})^m$ is $S^\star = [u, \dots, u]$.*

**Lemma 5** (Riemannian smoothness). *Let $\mathcal{M} = (\mathbb{S}^{d-1})^m$ and endow each block with the canonical metric. Then $\mathcal{L}_{\mathrm{KD}}$ is $\mathsf{L}_R$-smooth on $\mathcal{M}$ in the Riemannian sense: there exists a constant*

$$\mathsf{L}_R \le \frac{2\lambda}{m} + \frac{4\vartheta}{m}$$

*such that for all $S, S' \in \mathcal{M}$, $\|\operatorname{grad} \mathcal{L}_{\mathrm{KD}}(S) - \operatorname{grad} \mathcal{L}_{\mathrm{KD}}(S')\| \le \mathsf{L}_R \operatorname{dist}_{\mathcal{M}}(S, S')$. Sketch. For each block $o_i$, $\nabla_{o_i} \mathcal{L}_{\cos} = -(\lambda/m)u$ (constant), and $\nabla_{o_i} \mathcal{L}_{\mathrm{var}} = (2\vartheta/m)(o_i - \mu)$ with $\mu$ depending linearly on $\{o_j\}$. Projecting to the tangent space by $(I - o_i o_i^\top)$ and using the Lipschitzness of the projection map on $\mathbb{S}^{d-1}$ yields the bound.*

**Theorem 4** (Convergence of cosine+variance KD on the sphere). *Consider Riemannian gradient descent on $\mathcal{M} = (\mathbb{S}^{d-1})^m$:*

$$o_i^{(t+1)} = R_{o_i^{(t)}}\big(-\eta_t \operatorname{grad}_{o_i} \mathcal{L}_{\mathrm{KD}}(S_t)\big) \quad (i = 1, \dots, m),$$

*with the retraction $R_o(v) = (o + v)/\|o + v\|$. If the step sizes satisfy either (a) a constant stepsize $0 < \eta_t < 2/\mathsf{L}_R$, or (b) diminishing stepsizes $\sum_t \eta_t = \infty$, $\sum_t \eta_t^2 < \infty$, then:*

$$\mathcal{L}_{\mathrm{KD}}(S_t) \downarrow \mathcal{L}_{\mathrm{KD}}(S^\star), \qquad \|\operatorname{grad} \mathcal{L}_{\mathrm{KD}}(S_t)\| \to 0,$$

*and every limit point of $\{S_t\}$ is a Riemannian critical point. By Lemma 4, the unique global minimizer is $S^\star = [u, \dots, u]$; thus the sequence converges to $S^\star$.*

*Proof sketch.* Riemannian smoothness (Lemma 5) on the compact manifold $\mathcal{M}$ ensures the standard descent lemma and monotone decrease for RGD under $0 < \eta < 2/\mathsf{L}_R$, implying convergence of function values and gradients to zero. By Lemmas 3–4, the only global minimizer is $S^\star$, hence all limit points coincide with $S^\star$. $\square$

## D.5 Stability of Masked Gradient Updates

Let $g_i = \nabla_{W'_{v,i}} \mathcal{L} \in \mathbb{R}^d$. A coordinate mask $M_i \in \{0,1\}^d$ acts by $(M_i \odot g_i)_p = M_i(p)\, g_{i,p}$.

**Lemma 6** (Norm Bound under Masked Updates). *For any stepsize $\eta > 0$, the masked update $\Delta W'_{v,i} = -\eta(M_i \odot g_i)$ satisfies*

$$\|\Delta W'_{v,i}\|_2 \leq \eta \, \|g_i\|_2.$$

*Proof.* Coordinate-wise, $|M_i(p)\, g_{i,p}| \leq |g_{i,p}|$ because $M_i(p) \in \{0,1\}$. Hence $\|M_i \odot g_i\|_2 \leq \|g_i\|_2$, and multiplying by $\eta$ yields the claim. $\square$

**Theorem 5** (Inter-Shard Inner-Product Scaling). *Suppose that for each coordinate $p$, the masks $M_i(p), M_j(p) \in \{0,1\}$ are sampled independently with*

$$\Pr[M_i(p) = 1] = \Pr[M_j(p) = 1] = \rho, \quad 0 \leq \rho \leq 1,$$

*and masks are independent across coordinates and independent of $g_i, g_j$. Then, conditional on $g_i, g_j$,*

$$\mathbb{E}[\langle M_i \odot g_i,\, M_j \odot g_j \rangle \,|\, g_i, g_j] = \rho^2 \, \langle g_i, g_j \rangle.$$

*In particular, masking scales the expected cross-shard alignment/conflict by the factor $\rho^2$.*

*Proof.* By linearity of expectation and independence, for each coordinate $p$, $\mathbb{E}[M_i(p)M_j(p)] = \mathbb{E}[M_i(p)]\,\mathbb{E}[M_j(p)] = \rho^2$. Summing over $p$ yields the result. $\square$

## D.6 Closed-Loop Re-trigger Analysis

**Assumption 3** (Error Reduction per Re-trigger). *Let $r_i^{(n)}$ denote the error rate of shard $i$ after $n$ re-triggers. There exists $\delta > 0$ such that each re-trigger reduces error by at least $\delta$ whenever $r_i^{(n)} > \tau_{\mathrm{prune}}$.*

**Lemma 7** (Piecewise-Linear Error Decrease). *Under Assumption 3, for all $n \geq 0$,*

$$r_i^{(n)} \leq \max\{\, \tau_{\mathrm{prune}},\, r_i^{(0)} - n\delta \,\}.$$

*Proof.* If $r_i^{(k)} > \tau_{\mathrm{prune}}$, then $r_i^{(k+1)} \leq r_i^{(k)} - \delta$. Once $r_i^{(k)} \leq \tau_{\mathrm{prune}}$, the bound $r_i^{(n)} \leq \tau_{\mathrm{prune}}$ propagates for all $n \geq k$. Unrolling gives the stated maximum form. $\square$

**Theorem 6** (Finite-Time Hitting the Pruning Threshold). *Let*

$$N_\star = \left\lceil \frac{(r_i^{(0)} - \tau_{\mathrm{prune}})_+}{\delta} \right\rceil \quad \text{where } (x)_+ := \max\{x, 0\}.$$

*After at most $N_\star$ re-triggers, we have $r_i^{(N_\star)} \leq \tau_{\mathrm{prune}}$.*

*Proof.* By Lemma 7, choose the smallest integer $N_\star$ such that $r_i^{(0)} - N_\star\delta \leq \tau_{\mathrm{prune}}$. Then $r_i^{(N_\star)} \leq \tau_{\mathrm{prune}}$. $\square$

# E Algorithms

The pseudocode for error feedback, network pruning, sample knowledge distillation and reintegration, and the loss-based weighted ties merge strategy is as follows:

---

**Algorithm 1** REPAIR: Closed-Loop Lifelong Model Editing (Training)

---

**Require:** Pretrained model $f_{\theta_0}$; target FFN value matrix $W_v$; #shards $K$; mask ratio $\rho$; thresholds $(\epsilon, \tau_E, \tau_{\text{prune}}, \tau_{\text{correct}}, \epsilon_{\text{cons}})$; margins $(\gamma_1, \gamma_2, \gamma)$; KD weights $(\lambda, \vartheta)$ for Eq.(4); routing-loss weight $\lambda_a$; batch size $b$; optional temperature $T$ for soft KD.

1: Initialize side memories $W'_{v,i} \leftarrow W_v$ and masks $M_i \sim \text{Bernoulli}(\rho)$ for $i = 1..K$; feedback pool $\mathcal{E} \leftarrow \emptyset$; residual pool $\mathcal{R} \leftarrow \emptyset$.

2: **for** each incoming edit triple $(x_e, y_e, x_{\text{loc}})$ **do**

3:     $i^\star \leftarrow \text{AssignShard}(x_e)$                         ▷ Shard assignment by activation score

4:     $\mathcal{B} \leftarrow \text{FormBatches}(\{x_e\} \cup \mathcal{R}, b)$             ▷ Distribution-aware batching

5:     **for** each batch $B = \{x^{(0)}, \dots, x^{(b-1)}\} \in \mathcal{B}$ **do**

6:        $i \leftarrow \text{AssignShard}(x^{(0)})$                     ▷ Target shard for this batch

7:        $L_{\text{edit}} \leftarrow \text{AutoregCE}(B)$                   ▷ Autoregressive cross-entropy

8:        $L_{\text{KD}} \leftarrow \text{IntraBatchKD}(B, \lambda, \vartheta, T)$        ▷ Eq.(4); optional soft KD

9:        $L_{\text{act}} \leftarrow \text{RoutingMargin}(B, \gamma_1, \gamma_2, \gamma)$                ▷ Eq.(7)

10:       $L_{\text{batch}} \leftarrow L_{\text{edit}} + \lambda_a L_{\text{act}} + L_{\text{KD}}$

11:       $\text{MaskedUpdate}(W'_{v,i}, M_i, L_{\text{batch}})$     ▷ $W'_{v,i} \leftarrow W'_{v,i} - \eta(M_i \odot \nabla L)$

12:       $\text{FilterAndRecluster}(B, \epsilon_{\text{cons}}, \mathcal{R})$    ▷ Move high-$L_{\text{KD}}$ samples to residual pool

13:     **end for**

14:     $(\hat{y}, c) \leftarrow \text{Evaluate}(x_e, y_e)$               ▷ $c \in \{0, 1\}$ indicates success

15:     **if** $c = 0$ **then**

16:       $\mathcal{E} \leftarrow \mathcal{E} \cup \{(x_e, y_e)\}$

17:     **end if**

18:     **if** $|\mathcal{E}| > \tau_E$ **or** $\max_i \text{ErrorRate}(\mathcal{E}, i) > \tau_{\text{prune}}$ **then**

19:       $\text{ReTrigger}(\mathcal{E})$                        ▷ Prune worst shard, rebuild, and retrain

20:     **end if**

21: **end for**

22: $\text{LossAwareTIESMerge}(\{W'_{v,i}\}_{i=1}^K, W_v)$       ▷ Loss-aware weighted TIES merge

---

**Algorithm 2** REPAIR Inference with Dual-Memory Routing

---

1: **function** $\text{RouteAndPredict}(x)$

2:     compute $a(x) \leftarrow \text{FFNActivation}(x)$        ▷ Activation $a(x)$ at the target FFN layer

3:     **for** $i = 1..K$ **do**

4:       $\Delta_{\text{act}}^{(i)}(x) \leftarrow \| a(x) \cdot (W'_{v,i} - W_v) \|_2$

5:     **end for**

6:     **if** $\max_i \Delta_{\text{act}}^{(i)}(x) \leq \epsilon$ **then**

7:       **return** $f_{\theta_0}(x; W_v)$                      ▷ Route to main memory

8:     **else**

9:       $i^\star \leftarrow \arg\max_i \Delta_{\text{act}}^{(i)}(x)$

10:      **return** $f_{\theta_0}(x; W'_{v,i^\star})$                  ▷ Route to side memory $i^\star$

11:     **end if**

12: **end function**

---

**Algorithm 3** Training Subroutines

1: **function** ASSIGNSHARD($x$)
2:     $a \leftarrow$ FFNACTIVATION($x$);  $\Delta^{(i)} \leftarrow \|a \cdot (W'_{v,i} - W_v)\|_2, \ i = 1..K$
3:     **return** $\arg\max_i \Delta^{(i)}$              ▷ Use the most active shard during training
4: **end function**
5: **function** FORMBATCHES($S, \ b$)              ▷ Distribution-aware batching
6:     $o_i \leftarrow \mathrm{Norm}\big(\mathrm{ModelFeat}(x^{(i)})\big)$ for $i = 0, \ldots, b-1$
7:     Greedy seeding: pick $x^{(0)} = \arg\max_{x \in S} \frac{1}{|S|} \sum_{x'} \cos(o(x), o(x'))$
8:     Build $B \leftarrow \{x^{(0)}\} \cup \mathrm{Top}\text{-}(b-1)$ nearest by cosine; remove $B$ from $S$
9:     Repeat until $S$ is empty; **return** list of batches $\mathcal{B}$
10: **end function**
11: **function** AUTOREGCE($B$)            ▷ Autoregressive edit loss $L_{\mathrm{edit}}$
12:     $L \leftarrow 0$
13:     **for** $x \in B$ with target sequence $y$ **do**
14:         $L \leftarrow L - \sum_{t=1}^{|y|} \log p_\theta(y_t \mid y_{<t}, x)$
15:     **end for**
16:     **return** $L/|B|$
17: **end function**
18: **function** INTRABATCHKD($B, \lambda, \vartheta, T$)      ▷ Eq.(4); optional soft-KD
19:     Compute $o_i \leftarrow \mathrm{Norm}\big(\mathrm{ModelFeat}(x^{(i)})\big)$ for $i = 0, \ldots, b-1$
20:     $L_{\cos} \leftarrow \frac{1}{b-1} \sum_{i=1}^{b-1} \left(1 - \frac{o_i^\top o_0}{\|o_i\|\|o_0\|}\right)$
21:     $o_{\mathrm{mean}} \leftarrow \frac{1}{b} \sum_{i=0}^{b-1} o_i$;  $L_{\mathrm{var}} \leftarrow \frac{1}{b} \sum_{i=0}^{b-1} \|o_i - o_{\mathrm{mean}}\|_2^2$
22:     $L \leftarrow \lambda\, L_{\cos} + \vartheta\, L_{\mathrm{var}}$
23:     **if** $T > 0$ **then**        ▷ Optional: KL distillation for added stability
24:         Get logits $z_i$; $p_i = \mathrm{softmax}(z_i/T)$;  $L \leftarrow L + \frac{1}{b-1} \sum_{i=1}^{b-1} \mathrm{KL}(p_0 \| p_i)$
25:     **end if**
26:     **return** $L$
27: **end function**
28: **function** ROUTINGMARGIN($B, \gamma_1, \gamma_2, \gamma$)          ▷ Eq.(7)
29:     $L \leftarrow 0$
30:     **for** each edit sample $x_e \in B$ **do**
31:         sample unrelated $x_i$;  compute $\Delta_e = \mathrm{ACTDELTA}(x_e), \Delta_i = \mathrm{ACTDELTA}(x_i)$
32:         $L \leftarrow L + \max(0, \Delta_i - \gamma_1) + \max(0, \gamma_2 - \Delta_e) + \max(0, \gamma - (\Delta_e - \Delta_i))$
33:     **end for**
34:     **return** $L/|B|$
35: **end function**
36: **function** MASKEDUPDATE($W'_{v,i}, M_i, L$)     ▷ Masked gradient to reduce cross-shard interference
37:     $g \leftarrow \nabla_{W'_{v,i}} L$;  $g_{\mathrm{m}} \leftarrow M_i \odot g$         ▷ $M_i \in \{0,1\}^{\mathrm{shape}(W_v)}$
38:     $W'_{v,i} \leftarrow \mathrm{OptimizerStep}(W'_{v,i}, g_{\mathrm{m}})$         ▷ SGD/Adam, etc.
39: **end function**
40: **function** FILTERANDRECLUSTER($B, \epsilon_{\mathrm{cons}}, \mathcal{R}$)
41:     **for** $x \in B$ **do**
42:         $\ell_{\mathrm{KD}}(x) \leftarrow$ per-sample KD vs. $x^{(0)}$
43:         **if** $\ell_{\mathrm{KD}}(x) \geq \epsilon_{\mathrm{cons}}$ **then**
44:             move $x$ to $\mathcal{R}$
45:         **end if**
46:     **end for**
47:     **return**
48: **end function**
49: **function** EVALUATE($x_e, y_e$)
50:     $\hat{y} \leftarrow \mathrm{ROUTEANDPREDICT}(x_e)$;  $c \leftarrow \mathbf{1}[\hat{y} = y_e]$
51:     **return** $(\hat{y}, c)$
52: **end function**

**Algorithm 4** Utility Functions

1: **function** ERRORRATE($\mathcal{E}, i$)                                  ▷ Error rate for shard $i$
2:     $\mathcal{E}_i \leftarrow \{x \in \mathcal{E} \mid \arg\max_j \Delta_{\text{act}}^{(j)}(x) = i\}$
3:     $r_i \leftarrow \frac{|\{x \in \mathcal{E}_i | \text{CORRECTNESS}(x) \leq \tau_{\text{correct}}\}|}{|\mathcal{E}_i|}$
4:     **return** $r_i$
5: **end function**
6: **procedure** RETRIGGER($\mathcal{E}$)                                  ▷ Closed-loop pruning and retraining
7:     $j \leftarrow \arg\max_i \text{ERRORRATE}(\mathcal{E}, i)$                    ▷ Identify worst-performing shard
8:     Remove or reinitialize shard $j$: $W'_{v,j} \leftarrow W_v + \sigma_{\text{init}} \cdot \mathcal{N}(0, 1)$; resample $M_j$
9:     Build $\mathcal{E}_{\text{retrain}}$ from $\mathcal{E}$; form batches; retrain shards via MASKEDUPDATE + INTRABATCHKD
10: **end procedure**
11: **function** LOSSAWARETIESMERGE($\{W'_{v,i}\}, W_v$)              ▷ Loss-aware weighted TIES merge
12:     For each shard $i$: $\tau_i \leftarrow W'_{v,i} - W_v$;  compute training loss $L_i$ on its assigned data
13:     $w_i \leftarrow \frac{e^{-\alpha L_i}}{\sum_j e^{-\alpha L_j}}$
14:     **for** each parameter index $p$ **do**
15:         $S \leftarrow \{(i, \tau_i[p], w_i)\}_{i=1}^{K}$
16:         **if** all $\tau_i[p]$ share the same sign **then**
17:             $\delta[p] \leftarrow \sum_i w_i \tau_i[p]$                      ▷ Consistent signs: weighted sum
18:         **else**
19:             $i^\star \leftarrow \arg\max_i\{w_i |\tau_i[p]|\}$;  $\delta[p] \leftarrow \tau_{i^\star}[p]$  ▷ Conflict: keep most trustworthy shard
20:         **end if**
21:     **end for**
22:     $W_v \leftarrow W_v + \delta$;  **return** $W_v$
23: **end function**
24: **function** FFNACTIVATION($x$)
25:     **return** activation $a(x)$ at the target FFN layer
26: **end function**
27: **function** ACTDELTA($x$)
28:     **return** $\max_i \|a(x) \cdot (W'_{v,i} - W_v)\|_2$
29: **end function**
30: **function** MODELFEAT($x$)
31:     **return** feature used for similarity (e.g., $a(x)$ or last-token state)
32: **end function**
33: **function** NORM($v$)
34:     **return** $v/\|v\|_2$
35: **end function**
36: **function** CORRECTNESS($x$)
37:     **return** predicted correctness score for $x$
38: **end function**

