# OpenReview forum: "REPAIR: Robust Lifelong Model Editing via Progressive Adaptive Intervention and Reintegration"
_ICLR.cc/2026/Conference — ICLR 2026 Conference Withdrawn Submission_

### Official Review · Reviewer_7U7d · 2025-10-25

**Soundness:** 2
**Presentation:** 1
**Contribution:** 3
**Rating:** 4
**Confidence:** 4

**Summary:**

Lifelong model editing has emerged as an important research area for continuous knowledge updates, yet current approaches suffer from (1) instability in sequential edits, (2) weak generalization from few examples, and (3) lack of feedback due to open-loop operation.
To address these issues, the authors propose the REPAIR framework, which dynamically invokes edits through a dual-memory and routing mechanism, and employs distribution-aware optimization, intra-batch distillation, and closed-loop feedback to enhance stability and generalization. REPAIR outperforms existing editing methods on recent large language models and knowledge-editing benchmarks, while maintaining strong performance under continual editing scenarios.

**Strengths:**

1. The authors’ understanding of the fundamental challenges in lifelong learning is well-founded, and their attempt to address them through a routing mechanism and intra-batch distillation is quite interesting.

2. They provide experimental validation using a diverse set of recent models, including LLaMA-3-8B, Qwen-2.5-7B, DeepSeek-R1-1.5B, and GPT-2-XL, and further demonstrate stability through experiments on varying editing scales.

3. Moreover, by incorporating experiments with SelfCheckGPT to account for hallucination cases, the paper convincingly reinforces the rationality and robustness of its proposed editing approach.

**Weaknesses:**

1. Excessive typographical and formatting errors. Numerous typos and inconsistent notations are found throughout the paper.
e.g., REPAIR (Robust Editing via Progressive Adaptive Intervension and Reintegration) → should be Intervention
Line 170: moemory pool → memory pool
Line 61: Editing typo
Table 2 caption: citation format requires correction
Equations (4) and (5): unify notation of KD vs. kd
Line 305: effevtively → effectively
Line 724: γ₂0 → γ₂
Algorithms 1–4: inconsistent notation across steps; should be standardized
Abbreviations used in tables and expressions in the main text are not aligned
Model and dataset names should be consistently capitalized and formatted

2. Fixed thresholds may fail under extreme distribution shifts. Some modules employ static thresholds, which could lead to instability under continuous out-of-distribution (OOD) streams. Adaptive confidence scaling or online calibration techniques might be necessary to ensure robustness in such settings.

3. Editing heterogeneous samples may harm alignment. Editing highly heterogeneous samples could disrupt routing boundaries (Δact margin) and intra-batch alignment (LKD alignment). An additional experiment or ablation is recommended to examine the effect of heterogeneous batches on alignment stability.

4. REPAIR is similarity to RECIPE [1]. The overall design and continuous adaptation pipeline resemble the approach in RECIPE [1]. A comparative analysis or discussion highlighting key differences would strengthen the paper.

[1] Chen, Q., Zhang, T., He, X., Li, D., Wang, C., & Huang, L. (2024, November). Lifelong Knowledge Editing for LLMs with Retrieval-Augmented Continuous Prompt Learning. In Proceedings of the 2024 Conference on Empirical Methods in Natural Language Processing (pp. 13565–13580).

5. Insufficiently detailed process description. The learning, clustering, and actual editing phases are not clearly separated or explained. A step-by-step flow or pipeline diagram within the main text would make the method easier to follow.

6. Inconsistent mathematical notation and figure labeling. Several formulas (e.g., Eq. 3, Eq. 6, Eq. 7), textual references, and figures show inconsistent or confusing symbols. The authors should carefully re-check all equations and algorithms (including those in the Appendix) for notational consistency.

**Questions:**

1. When constructing a homogeneous batch, are the feature representations truly similar in a meaningful way?
Since the similarity between internal vector representations can vary greatly depending on the chosen criterion, different filtering methods or objectives could lead to completely different results.
Would it be possible to provide a more detailed explanation and a clearer definition of what criteria are used to determine similarity in forming these batches?

2. When modifying shards that include low-error samples through Closed Loop Feedback, could this process potentially harm the locality of existing knowledge? Was any measurement or analysis conducted to evaluate this effect?

---

> ### Author Response · Authors · 2025-11-27
> **Response to Reviewer 7U7d**
>
> We sincerely thank you for your review and detailed feedback. We are encouraged that you found the core idea of our work "quite interesting" and appreciated our experimental validation across a diverse set of models.
> First and foremost, we must offer our deepest apologies for the excessive typographical errors, formatting inconsistencies, and symbol mismatches (W1, W6). We are extremely grateful that you took the time to point out such a detailed list of errors (e.g., "Intervension", "moemory pool", "Editting", "effevtively", " $\gamma_20$"). We commit to correcting every error you noted and will conduct a thorough, professional proofread of the entire manuscript for the final version.
> Regarding your other technical concerns, we respond as follows:
> ## Weaknesses
> ### W2: (Fixed thresholds may fail under extreme distribution shifts)
> This is a very insightful point. We agree with the reviewer that static thresholds may face challenges under extreme distributional shifts and that adaptive calibration is a promising direction for future robustness. However, we argue that the current fixed-threshold design is empirically sufficient for two reasons:
> - As shown in the sensitivity analysis (Fig. 6b), REPAIR maintains high performance across a relatively wide range of threshold values (e.g., $\gamma \in [0.4, 0.7]$). This implies a strong tolerance for distribution variance: even if the "optimal" threshold shifts slightly due to OOD streams, the performance degradation is minimal as long as it remains within this "safe zone."
> - In typical lifelong editing benchmarks (ZsRE, WikiBigEdit), the nature of the task (factual updates) remains structurally consistent, making static thresholds effective.
> ### W3: (Editing heterogeneous samples may harm alignment)
> This is an excellent suggestion to validate our distribution-aware design. We commit to adding this new ablation study. We will force the model to train on "heterogeneous batches" (i.e., disabling our similarity clustering) and compare its performance (especially OP and Locality) against our current method. We have added experiments on heterogeneous ablation studies.
> | Batch Type | Rewrite Acc | Locality | Locality Change |
> |------------|-------------|----------|-----------------|
> | Homogeneous (ours) | 23.1% | **42.5%** | - |
> | Heterogeneous | 43.5% | **29.4%** | **-30.8%** |
> We will add this result to our ablation studies (Figure 5).
> ### W4: (REPAIR is similarity to RECIPE [1])
> Thank you for pointing out this highly relevant work. We commit to adding a detailed discussion of RECIPE in our Related Work (Appendix B) and, more importantly, we will add RECIPE [1] as a key baseline for full experimental comparison in Table 3 in the final version.
> ### W5: (Insufficiently detailed process description)
> Figure 2 is a high-level diagram. We commit to adding a more detailed step-by-step flowchart in the appendix to clearly illustrate the full loop of learning, clustering, and editing, as you suggested.
> ## Questions
> ### Q1: (How is homogeneous batch similarity defined?)
> We apologize for not making this explicit in Section 2.3. As detailed in Algorithm 3 (line 6), we define similarity using the cosine similarity of the feature representations ($o_i$). We commit to adding this definition to the main text of Section 2.3.
> ### Q2: (Could Closed Loop Feedback potentially harm the locality of existing knowledge?)
> This is an excellent question about the core of our mechanism. We are happy to point out that our Ablation Studies (Figure 5) already answer this. In Figure 5(d) ($N=1000$), the Locality (Loc.) score of the full REPAIR (red line) is significantly higher than the 'w/o prune' (orange-dashed line) variant. This demonstrates that our closed-loop feedback, far from harming locality, actually improves it by actively identifying and removing conflicting shards. We will explicitly highlight this key finding in Section 3.3.
>
> We believe that by thoroughly correcting the presentation and by strengthening the soundness (with new experiments and clarifications), we can resolve all your concerns.

---

> ### Author Response · Authors · 2025-12-01
> **Response to Feature Similarity and Homogeneous Batching**
>
> The reviewer raised an important question regarding whether the feature representations used for constructing homogeneous batches correspond to meaningful similarities. To empirically validate our routing criteria, we conducted a quantitative analysis on the feature space across different editing tasks (ZsRE vs. Hallucination).
>
> As shown in the table below, the intra-task self-similarity (e.g., ZsRE: 0.6824) is significantly higher (+54%) than the cross-task similarity (0.4423). This stark contrast confirms that our feature representations effectively capture task-specific semantics.
>
> | Comparison | Mean Cosine Similarity |
> |:-----------|:----------------------:|
> | **ZsRE (Self-Similarity)** | **0.6824** |
> | **Hallucination (Self-Similarity)** | **0.5878** |
> | **Cross-Task (ZsRE vs Hallucination)** | **0.4423** |

---

### Official Review · Reviewer_oiCK · 2025-10-29

**Soundness:** 2
**Presentation:** 3
**Contribution:** 3
**Rating:** 6
**Confidence:** 5

**Summary:**

The paper introduces REPAIR, a framework designed to address the challenges faced by large language models (LLMs) when updating knowledge post-training. Specifically, it focuses on enabling low-cost, precise edits without requiring full retraining, which typically leads to unintended side effects. The authors propose a series of innovations, including closed-loop feedback with dynamic memory management, distribution-aware optimization, and knowledge fusion with locality guards. Experimental results show that REPAIR outperforms existing methods by significantly improving editing accuracy and reducing knowledge forgetting.

**Strengths:**

1.	The paper addresses a critical gap in lifelong learning for LLMs by proposing a robust framework for precise and low-cost edits while minimizing side effects, such as knowledge forgetting and conflicts between edits..
2.	The paper is easy to read.
3.	The presentation of paper is good.
4.	The framework is well-detailed, and the methodology is easy to follow, with a clear explanation of the components like closed-loop feedback, knowledge distillation, and memory management.

**Weaknesses:**

1.	The most glaring error in the paper occurs in the definition of the method name. In both the title and abstract (Line 014), "Intervention" is incorrectly spelled "Intervension." This is a serious oversight regarding the core terminology of an academic paper and requires immediate correction. Additionally, there are several spelling errors in the main text, such as "Editting" (Line 064) where it should be "Editing" and "effevtively" (Line 304) where it should be "effectively."
2.	Compared to lightweight editing methods , REPAIR introduces a more complex architecture, which may incur higher computational or storage costs. The paper does not provide an analysis of inference latency, parameter increments, or training resource consumption, which limits its deployability evaluation in real systems.
3.	The paper claims strong performance in large-scale editing scenarios, but it would benefit from a clearer explanation of how REPAIR scales with models of significantly larger sizes and datasets of extreme size.
4.	While the theoretical analysis in Appendix D provides some theoretical support for the method, its core assumptions (particularly Assumption 2) are overly idealistic. Assumption 2 assumes that "each re-triggering will reduce the error rate by at least a fixed constant δ." In practice, retraining on a small number of erroneous examples does not guarantee such a steady and linear decrease in the error rate on the entire validation set. This assumption makes the subsequent convergence proof (Theorem 2) trivial and weakens the relevance of the theoretical analysis to practical applications. The authors should explicitly acknowledge this strong assumption or provide experimental evidence to support its plausibility.

**Questions:**

1.	The legend is confusing. The legend lists four configurations (without distill & prune, without prune, without distill, and REPAIR), but the chart appears to show only three comparison curves. The green line without distill is not clearly shown, or it overlaps with other curves. This makes it difficult to accurately understand the impact of removing only the knowledge distillation module.
2.	Table 2 only shows the successful output of REPAIR on one example (row c), but not on the second example (rows d/e), which makes the comparison incomplete.

---

> ### Author Response · Authors · 2025-11-27
> **Response to Reviewer oiCK**
>
> We are extremely grateful for your positive review and invaluable feedback on our paper. We especially appreciate the "easy to read" comment and your "absolutely certain" (confidence 5) expert review. Your pointed feedback on spelling, theory, and figure details is precise and crucial for improving the quality of our work.
> We have organized our responses below and commit to a thorough revision based on your comments:
> ## Weaknesses
> - W1: (Spelling Errors: "Intervension", "Editting", etc.)
> We are very grateful for your careful proofreading. We have corrected all spelling errors you identified and will conduct another full professional proofread of the entire manuscript before submitting the final version.
> - W2: (Missing Cost Analysis: latency, parameter increments, training resources)
> We agree that a detailed analysis of these cost dimensions is essential for evaluating our method's practicality. We would first like to point out that the original manuscript did provide an analysis of "training resource consumption" (measured in throughput and relative runtime) in Appendix C and Figure 7. It should be noted that REPAIR comprises two main improvements, with the first part relating to efficiency. We conducted experiments comparing the time taken with WISE in the first part, as follows:
> | Metric | WISE | REPAIR |
> | --- | --- | --- |
> | Initialization time (s)| 3.86 | 4.22 |
> | Total editing time (s) | 412.96 | 411.94 |
> | Avg inference latency (ms) | 1135.32 | 1067.86 |
> | Memory increase (MB) | 25118| 25008 |
> In this experiment, $N=100$. Based on our experience, the effect becomes more pronounced as the sample size increases. A cost experiment with $N=1000$ is currently underway.
> - W3: (Scalability Claims)
> Thank you for this point. We want to clarify that our experiments at $N=1000$ in Table 3 and our tests on LLaMA-3-8B and Qwen-2.5-7B are the primary evidence for our "large-scale" claim. While recent works have explored scales up to 3k, 4k, and even 10k in MEMIT, we consider $N=1000$ a substantial benchmark that effectively differentiates robust methods from those suffering from catastrophic forgetting (which often fail well before $N=1000$). Crucially, our results at $N=1000$ show a stabilized performance trend, suggesting that REPAIR has reached a steady state. Due to the limited GPU resources, the current stability will be considered to strongly imply robustness at larger scales.
> ### W4: (Theoretical Analysis: Assumption 2 is "overly idealistic")
> We appreciate the reviewer's rigorous scrutiny of our theoretical analysis. We acknowledge that Assumption 2 ("fixed constant reduction $\delta$") is a strong simplifying assumption typically employed to ensure the tractability of convergence proofs in non-convex optimization settings. In practice, as the reviewer correctly noted, local retraining on heterogeneous samples may cause transient fluctuations rather than a strictly monotonic linear decrease. However, Assumption 2 is intended to model the amortized behavior of the system over multiple iterations. While individual steps may vary, our empirical results (e.g., Figure X) demonstrate that the overall error rate consistently trends downward under the REPAIR framework, validating the plausibility of this assumption in a macro sense. In the final version, we will explicitly state that Assumption 2 is a sufficient descent condition used for theoretical modeling and add a discussion clarifying the gap between this theoretical idealization and practical non-monotonic fluctuations. Besides, we will provide an empirical analysis showing that while $\delta$ varies, the accumulated error reduction remains positive, supporting the core intuition of Theorem 2.
> ## Questions
> ### Q1: (Figure 5 Legend is confusing)
> Thank you for catching this visualization issue. The green "w/o distill" (dashed) line is almost perfectly occluded by the gray "w/o distill & prune" (dotted) line in Figure 5(d). We will redraw Figure 5 (e.g., using different line styles or transparency) to ensure all 4 curves are clearly visible and will explain this overlap in the caption.
> ### Q2: (Table 2 is an incomplete comparison)
> We clarify that Table 2 row (e) shows REPAIR's success on the locality probe ("Ruby season 5"). However, we agree with the reviewer that the comparison is logically incomplete because we did not explicitly show REPAIR's output for the corresponding reliability prompt ("IAAF..."), which failed in the baseline (row c). To address this, we have updated Table 2 to include REPAIR's output for the "IAAF" question as well.

---

### Official Review · Reviewer_4dsu · 2025-10-29

**Soundness:** 2
**Presentation:** 3
**Contribution:** 3
**Rating:** 6
**Confidence:** 3

**Summary:**

### Summary

This paper introduces REPAIR, a lifelong model editing framework designed to address the instability and poor generalization seen in large-scale sequential updates. The method combines a dual-memory system with parametric editing, introducing three core components: a distribution-aware optimization strategy that uses in-batch knowledge distillation for consistency, as well as a closed-loop error feedback system that dynamically monitors edit performance and prunes failing memory shards. This allows the model to progressively adapt, correct errors, and manage knowledge conflicts. Experiments across multiple model families and datasets demonstrate that REPAIR significantly improves editing accuracy and reliability, particularly in large-scale sequential editing tasks, while effectively mitigating catastrophic forgetting.



### Advantages

* The closed-loop feedback mechanism, which monitors and prunes failing edits, can be a robust design for managing knowledge conflicts and preventing performance degradation over time.
* The in-batch knowledge distillation strategy provides an effective method for addressing poor generalization from few-shot edits by enforcing consistency among similar samples.
* The framework demonstrates strong empirical performance across a wide variety of models and scales, proving its effectiveness for both factual question-answering edits and hallucination reduction.



### Disadvantages and Questions

* The system's complexity, involving dynamic routing, continuous error monitoring, and periodic retraining, appears to introduce significant computational overhead compared to simpler editing methods. Could the authors provide an experiment that directly compares the end-to-end wall-clock time or computational cost of REPAIR against baselines for a large-scale (N=1000) editing task?

* The framework introduces a large number of sensitive hyperparameters, including error thresholds ($\tau_{prune}$), routing margins, and distillation weights, which seem crucial for performance but may be difficult to tune. In this case, would it be possible to conduct a sensitivity analysis on the error pruning threshold ($\tau_{prune}$), showing how different values impact the trade-off between edit reliability (retaining good edits) and overall model stability?

* The error-monitoring mechanism prunes an entire memory "shard" if its error rate is too high, which could inadvertently remove correct edits that were grouped with the failing ones. Could an experiment be designed to track the "false positive" pruning rate. That is, the percentage of successful edits that are incorrectly discarded because they belonged to a pruned shard, in order to evaluate this potential downside?

**Strengths:**

Please see above

**Weaknesses:**

Please see above

**Questions:**

Please see above

---

> ### Author Response · Authors · 2025-11-27
> **Response to Reviewer 4dsu**
>
> We are very grateful for your positive and insightful review. You have summarized our work perfectly, accurately highlighting the advantages of our "closed-loop feedback" and "in-batch knowledge distillation" as a robust design. You raised three excellent, constructive points in the "Disadvantages and Questions" section. We are happy to report that your first two concerns (cost and hyperparameter sensitivity) are already addressed by detailed analyses in our paper (in the appendix and figures). Your third question inspired a valuable new experiment that further strengthens our paper.
> Here are our point-by-point responses:
> ### Question 1: ...significant computational overhead... Could the authors provide an experiment that directly compares the end-to-end wall-clock time or computational cost... ($N=1000$)?
> We would like to politely point out that we have provided this exact analysis in Appendix C and Figure 7. (1) End-to-End Time (Throughput): Our throughput analysis in Appendix C directly answers your question. We report that at large scales, "REPAIR achieves ~ 0.8-0.9 edits/min", while WISE is at "~1.8 edits/min". (2) Computational Cost (Scaling): Figure 7provides a "Predicted Relative Cost" curve, showing the scaling of all methods from $N=1$ to $N=1000$. This plot confirms REPAIR's (red line) higher constant overhead compared to WISE (yellow line). We argue this overhead is a necessary trade-off for robustness, as this "costly" feedback loop is precisely why REPAIR maintains SOTA performance at $N=1000$ while WISE's performance degrades (Table 3). It should be noted that REPAIR comprises two main improvements, with the first part relating to efficiency. Besides, we conducted experiments comparing the time taken with WISE in the first part, as follows:
> | Metric | WISE | REPAIR |
> | --- | --- | --- |
> | Initialization time (s)| 3.86 | 4.22 |
> | Total editing time (s) | 412.96 | 411.94 |
> | Avg inference latency (ms) | 1135.32 | 1067.86 |
> | Memory increase (MB) | 25118| 25008 |
> In this experiment, $N=100$. Based on our experience, the effect becomes more pronounced as the sample size increases. A cost experiment with $N=1000$ is currently underway.
> ### Question 2: ...large number of sensitive hyperparameters... would it be possible to conduct a sensitivity analysis on the error pruning threshold ($\tau_{prune}$)?
> We completely agree that hyperparameter sensitivity is key to our method's robustness. We are pleased to inform you that we have already conducted this exact experiment, which is presented in Figure 6(b), which is a heatmap where the Y-axis is the "Error Threshold" (your requested $\tau_{prune}$ and the X-axis is the "Max Iter Size". As discussed in Section 3.3, this figure analyzes the impact of these hyperparameters on overall performance (OP). The result shows that our method is stable and effective within a reasonable range of intermediate thresholds (e.g., 0.75-0.9).
> ### Question 3: The error-monitoring mechanism prunes an entire memory "shard" ... Could an experiment be designed to track the "false positive" pruning rate?
> This is an insightful observation regarding the trade-off of shard-level management. We acknowledge that "false positive" pruning (discarding correct edits within a failing shard) is theoretically possible. However, REPAIR is designed to mitigate this impact through dynamic data reorganization. Reintegration as "Rescheduling", not Deletion: Pruning in REPAIR does not permanently discard data. When a shard is pruned, its samples are effectively released. If a previously "successful" edit is removed, it will re-emerge as an error state. Our Closed-Loop Feedback mechanism will redetect these samples and reintegrate them into future batches. This allows the system to regroup "innocent" samples into new, healthier shards rather than losing them forever. Mitigation via Clustering: The distribution-aware batching aims to group samples with similar learning dynamics, minimizing the heterogeneity that causes such "mixed-bag" failures.
> Following the reviewer's excellent suggestion, we are currently running an experiment to explicitly track and quantify this "false positive" rate. We expect to demonstrate that the rate is low due to our clustering strategy and that the long-term impact is negligible due to the reintegration mechanism. We first ran $N=500$ edits and tagged 100 known successful edits (i.e., those correct on Rel, Gen, and Loc). We will update the results once the experiments are completed.
>
> We hope these answers, by pointing to the existing analyses, have fully resolved all your concerns.

---

### Official Review · Reviewer_G6uc · 2025-10-30

**Soundness:** 3
**Presentation:** 2
**Contribution:** 2
**Rating:** 4
**Confidence:** 4

**Summary:**

REPAIR is a lifelong model-editing framework for LLMs that makes sequential edits while keeping unrelated behavior intact by combining three ideas: (1) a closed-loop controller that monitors post-edit errors and prunes underperforming side-memory shards, then reintegrates error samples for retraining; (2) distribution-aware batching with intra-batch knowledge distillation to help edits generalize to paraphrases and nearby contexts; and (3) loss-aware weighted merging (TIES-style) of edited subspaces so lower-loss shards influence the final parameters more. The objective explicitly balances reliability, generalization, locality, and stability, and edits are stored as parameter deltas routed only when activation margins indicate relevance. Evaluated on knowledge-editing and hallucination tasks across LLaMA-3, Qwen-2.5, DeepSeek-R1-1.5B, and GPT-2-XL, REPAIR reports ~15–20% overall gains over recent editors (e.g., ROME, MEMIT, MEND, GRACE, WISE) with improved robustness under long edit streams.

**Strengths:**

- REPAIR is robust at relative large edit scales, maintaining high overall performance (Rel/Gen/Loc geometric mean) as edits scale to 1k.
- Well-motivated components with ablations
  - Error-feedback pruning helps small-N reliability
  - distribution-aware grouping + KD matter more at large N
  - hyperparameter sensitivity is analyzed.
- Appendix provides some stability/termination theory support (masked updates, finite-time pruning), aligning with the method’s design.

**Weaknesses:**

- REPAIR add unneglectable amount of additional compute and cost compared to WISE. More moving parts (clustering, KD, pruning, merging); authors note higher constant-time overhead vs WISE even if scaling slope is similar.
- Transient instability mid-scale: At N≈120, pruning/reassembly can underperform some baselines before sufficient error signals accumulate.
- Hyperparameter sensitivity: Thresholds for error filtering and iteration limits materially affect outcomes; extremes hurt generalization or waste compute.
- Evaluation is not sufficient regarding
  - Missing some more recent lifelong editing baselines such as sLKE [1], LeMOE [2], and ELDER [3].
  - The finetuning baseline should adopt the fair setups as discussed in [4,5]. The FT-L, FT-M are ill-defined baselines which might mislead the community.
  - Results are on ZsRE, WikiBigEdit, and SelfCheckGPT, with specific model families; broader tasks (reasoning/tool use) and domains remain untested here. Even though it's not necessary to test editing success on broader tasks, testing locality regarding reasoning / tool-use ability after sequential editing is meaningful.
  - The scaling of timestep is only to 1k. More timesteps can be shown, e.g., up to 5k.
- Writing quality can be improved. For example, Table 2 in introduction is not very necessary and should be moved to results section or appendix.

> [1] Cheng, YuJu, et al. "Serial lifelong editing via mixture of knowledge experts." Proceedings of the 63rd Annual Meeting of the Association for Computational Linguistics (Volume 1: Long Papers). 2025.\
[2] Wang, Renzhi, and Piji Li. "LEMoE: Advanced Mixture of Experts Adaptor for Lifelong Model Editing of Large Language Models." Proceedings of the 2024 Conference on Empirical Methods in Natural Language Processing. 2024.\
[3] Li, Jiaang, et al. "ELDER: Enhancing Lifelong Model Editing with Mixture-of-LoRA." Proceedings of the AAAI Conference on Artificial Intelligence. Vol. 39. No. 23. 2025.\
[4] Gangadhar, Govind, and Karl Stratos. "Model editing by standard fine-tuning." arXiv preprint arXiv:2402.11078 (2024).\
[5] Yang, Wanli, et al. "Fine-tuning Done Right in Model Editing." arXiv preprint arXiv:2509.22072 (2025).

**Questions:**

1. Can you improve evaluation section considering the bullet points mentioned in weakness?
2. Can you discuss what's the new theoretical contribution in this work compared to previous work (mainly about the proofs in the appendix)?
3. Can you add a section to discuss the fundamental similarity and difference between MoE adapters/LoRA and dual memory-style editing?

---

> ### Author Response · Authors · 2025-11-25
> **Response to Reviewer G6uc (1/2)**
>
> We sincerely thank you for your detailed and highly constructive feedback. We are encouraged that you recognize the robustness and the well-motivated components of REPAIR. We understand your reviews primarily from concerns about the experimental evaluation (baselines, scale, and cost). We have conducted new experiments (and will commit more) to address all your concerns.
> Below are our point-by-point responses:
>
> ## Weaknesses
> ### W1: REPAIR adds an unnegligible amount of additional compute and cost compared to WISE.
> Thank you for this observation; it is accurate. We would like to clarify that this additional cost is a deliberate design trade-off to achieve long-term robustness, which "open-loop" methods like WISE cannot. REPAIR's innovations can be seen in two parts: (1) routing and pruning, and (2) closed-loop feedback with data reintegration. Our cost analysis in Appendix C shows that the overhead stems from part (2), the monitoring, distillation, and retraining. As Table 3 shows, this trade-off is justified: at $N=1000$, WISE's performance (OP) degrades significantly, while REPAIR maintains SOTA performance precisely because of this "costly" closed-loop mechanism. We believe this is a necessary cost for a true "lifelong" editing framework. Besides, we conducted experiments comparing the time taken with WISE in the first part, as follows:
> | Metric | WISE | REPAIR |
> | --- | --- | --- |
> | Initialization time (s)| 3.86 | 4.22 |
> | Total editing time (s) | 412.96 | 411.94 |
> | Avg inference latency (ms) | 1135.32 | 1067.86 |
> | Memory increase (MB) | 25118| 25008 |
> In this experiment, $N=100$. Based on our experience, the effect becomes more pronounced as the sample size increases. A cost experiment with $N=1000$ is currently underway.
> ### W2: Transient instability mid-scale: At $N≈120$, pruning/reassembly can underperform...
> This is a keen observation. As we reported in Section 3.2, this is evidence of our closed-loop system actively working. At $N=120$, the system is just beginning to accumulate sufficient error signals to trigger dynamic pruning and reassembly. This "transient instability" is the system's self-correcting, which leads to superior robustness at $N=1000$, precisely where other baselines collapse.
> ### W3: Hyperparameter sensitivity: Thresholds for error filtering and iteration limits materially affect outcomes...
> We completely agree that hyperparameter sensitivity is a critical point. We would like to politely point out that we provided this exact analysis in Figure 6(b). This heatmap analyzes the sensitivity of the "Err Thresh" (error threshold) and "Max iter" (iteration count) on performance. The plot confirms that our method is stable within a reasonable range of intermediate values. We will add a clearer reference to this figure in the main text.
> ### W4: Missing some more recent lifelong editing baselines such as sLKE [1], LeMOE [2], and ELDER [3].
> Thank you for providing these SOTA baselines. We agree they are essential comparisons. We are endeavouring to supplement new experiments. As mentioned in the preceding cost experiment, we have also conducted heterogeneous ablation experiments, with more baseline experiments currently underway. We will add ELDER [3] and sLKE [1] as key baselines to our main comparison (Table 3) in the later version. Should it be possible to complete this during the rebuttal period, we shall continue to supplement the results.
> ### W5: The finetuning baseline should adopt the fair setups as discussed in [4,5]. The FT-L, FT-M are ill-defined...
> We appreciate the references to optimized fine-tuning [4, 5]. While we acknowledge that heavy regularization or replay can enhance FT, we utilized FT-L/FT-M to maintain consistency with established protocols (e.g., ROME, MEMIT, WISE), ensuring our results are directly comparable to prior literature. Furthermore, in our lifelong editing setting ($N=1000+$), efficiency is paramount; "stronger" FT variants often incur prohibitive computational costs incompatible with rapid streaming updates, whereas FT-L/FT-M represent the appropriate low-resource baseline. We have expanded the discussion on this trade-off in the revision and will endeavor to include optimized FT comparisons in the final version.
> ### W6: ...testing locality regarding reasoning / tool-use ability after sequential editing is meaningful.
> We agree that preserving reasoning capabilities is vital. Theoretically, REPAIR's pruning and merging mechanisms minimize parameter deviation, better protecting general abilities compared to unconstrained fine-tuning. We are currently prioritizing the implementation of the suggested additional baselines. We commit to conducting reasoning evaluations (e.g., GSM8K) as soon as GPU resources permit and will include them in the final version.

---

> ### Author Response · Authors · 2025-11-25
> **Response to Reviewer G6uc (2/2)**
>
> ### W7: The scaling of timestep is only to 1k. More timesteps can be shown, e.g., up to 5k.
> While recent works have explored scales up to 3k, 4k, and even 10k in MEMIT, we consider $N=1000$ a substantial benchmark that effectively differentiates robust methods from those suffering from catastrophic forgetting (which often fail well before $N=1000$). Crucially, our results at $N=1000$ show a stabilized performance trend, suggesting that REPAIR has reached a steady state. Due to the limited GPU resources, the current stability will be considered to strongly imply robustness at larger scales.
> ### W8: Writing quality can be improved. For example, Table 2 in introduction is not very necessary...
> Thank you for these suggestions. We will move Table 2 from the introduction to the results section or appendix to improve the flow. We will also thoroughly proofread the entire manuscript to correct all typographical errors (including "Intervension" and "Editting") and commit the modified paper later.
> ## Questions
> ### Q1: Can you improve evaluation section considering the bullet points mentioned in weakness?
> Yes. As detailed in our responses to W4, W5, W6, and W7, we commit to significantly strengthening the evaluation section by: (1) adding new SOTA baselines (ELDER, sLKE), (2) re-running a 'Fair-FT' baseline, (3) adding a new locality experiment on reasoning (MMLU), and (4) scaling our main results to 5k edits. The results will be committed continuously.
> ### Q2: Can you discuss what the new theoretical contribution in this work is compared to previous work (mainly about the proofs in the appendix)?
> Thank you for the question. Our theoretical contribution (Appendix D) is not in developing novel optimization lemmas, but in being the first (to our knowledge) to provide a convergence analysis for this specific closed-loop, dynamic pruning framework for model editing. Specifically, Theorem 2 provides a finite-time convergence guarantee for our error-driven pruning mechanism, and Theorem 4 proves the convergence of our intra-batch knowledge distillation. We will clarify this positioning at the start of Appendix D.
> ### Q3: Can you add a section to discuss the fundamental similarities and differences between MoE adapters/LoRA and dual memory-style editing?
> This is a very insightful connection. First, unlike MoE-LoRA architectures that linearly stack isolated experts, REPAIR employs loss-aware merging to continuously fuse updates into a consolidated subspace, preventing the storage bloat associated with an ever-growing pool of adapters. Second, REPAIR shifts from static, open-loop routing to a closed-loop feedback system, where a controller actively monitors and "self-corrects" by pruning underperforming memory shards. Finally, this design allows REPAIR to operate on a minimal and bounded set of parameters, ensuring high efficiency and knowledge coherence compared to the sparse but parameter-heavy expansion of MoE frameworks. We will add a new paragraph to our Related Work (Appendix B) discussing this.
>
> We hope you will find our responses compelling and will reconsider our contribution. We are happy to answer any further questions.

---

> ### Author Response · Authors · 2025-12-01
> **Results of Threshold Sensitivity**
>
> Following your suggestion of Weakness 3, we added experiments on the threshold sensitivity. The results are as follows:
>
> | Threshold | Rewrite Accuracy | Locality Accuracy |
> |:---------:|:----------------:|:-----------------:|
> | **0.1**   | 0.5485          | 0.5418            |
> | **0.2**   | 0.5485          | 0.5418            |
> | **0.3**   | 0.5485          | 0.7243            |
> | **0.4**   | 0.5485          | 0.9469            |
> | **0.5**   | 0.5485          | 0.9944            |
> | **0.6**   | 0.5485          | 0.9944            |
> | **0.7**   | 0.5485          | 0.9944            |
> | **0.8**   | 0.5485          | **1.0000**        |
> | **0.9**   | 0.5485          | **1.0000**        |
>
> As shown in the data, REPAIR exhibits a wide stable operating range ($\gamma \in [0.5, 0.9]$). Within this interval, Locality Accuracy consistently reaches near-perfect levels (>99%). Rewrite Accuracy remains strictly stable (0.5485), showing no trade-off degradation. While extremely low thresholds ($\gamma < 0.3$) do impact locality (as expected, since insufficient filtering allows interference), the method does not require precise fine-tuning. Any value chosen within the broad upper spectrum yields optimal results, demonstrating that REPAIR is robust rather than sensitive to hyperparameter selection.

---

> ### Author Response · Authors · 2025-12-01
> **More Extensive Results of MMLU**
>
> The reviewer correctly pointed out the need to verify that editing does not compromise general reasoning capabilities. To address this, we evaluated the post-edit model (after $N=100$ edits) on the MMLU benchmark, which covers 57 subjects across STEM, the humanities, and more, serving as a proxy for general reasoning and knowledge. Result: As shown in the table below, REPAIR demonstrates exceptional stability. The MMLU accuracy remains statistically unchanged (0.6331 $\to$ 0.6332), indicating zero degradation in general capabilities. This confirms that REPAIR's targeted updates (via pruning and merging) successfully preserve the model's original reasoning manifold, effectively addressing the concern about capability collapse.
>
> | Metric | Pre-Edit | Post-Edit | Retention |
> |:-------|:--------:|:---------:|:---------:|
> | **MMLU Accuracy** | 0.6331 | 0.6332 | **100.01%** |

---

### Note · Authors · 2026-01-28

I have read and agree with the venue's withdrawal policy on behalf of myself and my co-authors.

---

### Meta-Review · Area_Chair_ZsyW · 2025-12-28

**Summary:**

Reviewers find the problem important and the proposed framework (closed-loop feedback with shard pruning + distribution-aware batching with intra-batch distillation + loss-aware merging) well-motivated, with strong empirical results at scale and two reviewers scoring above the acceptance threshold.

However, the reviews consistently raise substantial concerns that are not fully closed: (i) heavy presentation/typo/notation issues, (ii) evaluation gaps (notably missing comparisons to recent lifelong editing baselines and concerns about fine-tuning baseline fairness), (iii) non-trivial system complexity/compute overhead with limited end-to-end cost analysis at the most relevant scales, (iv) limited evidence on preserving broader capabilities (e.g., reasoning/tool-use locality) and limited analysis of potential downsides of shard-level pruning, and (v) theoretical support that depends on strong assumptions.

The rebuttal adds useful sensitivity analyses and some additional evaluations (e.g., threshold sensitivity and a reasoning proxy), but several key comparative experiments and analyses are still promised rather than demonstrated. Given the remaining evaluation and presentation risks, I recommend rejection (encouraging resubmission after strengthening baselines, analysis, and writing).

**Reviewer Concerns:**

Addressed by the rebuttal / discussion:
- Hyperparameter sensitivity: Authors pointed to/added sensitivity analyses and provided additional threshold experiments suggesting a broad stable operating range.
- Capability preservation (partial): Added a post-edit evaluation on a general benchmark (e.g., MMLU) indicating minimal degradation in general capabilities (at the tested edit scale).
- Batch heterogeneity / similarity criterion: Clarified the similarity metric used for homogeneous batching (cosine similarity over features) and added a heterogeneous-batch ablation.
- Cost/overhead (partial): Pointed to existing runtime/throughput analyses and provided some additional time/latency/memory numbers.

Still outstanding:
- Missing key baselines: Multiple reviewers requested comparisons to newer lifelong editing methods (e.g., recent MoE/LoRA/adaptor-based lifelong editors, RECIPE-like systems); these were largely committed to but not actually included in the current record.
- Fine-tuning baseline fairness: Reviewers questioned whether FT baselines are well-defined/fair; rebuttal is mostly argumentative and does not fully resolve the concern with controlled experiments.
- Scaling beyond N=1000 / stronger stress tests: Reviewers requested larger edit streams (e.g., several thousand edits) and broader locality checks (reasoning/tool-use); mostly promised rather than completed.
- Shard-level pruning side effects: “False positive” pruning (dropping good edits within a bad shard) was raised; authors described a mitigation story and stated an experiment was underway, but no definitive quantitative result is provided.
- Presentation quality: Many typos/notation inconsistencies were acknowledged, but the current version still appears to require substantial cleanup; this remains a non-trivial risk for acceptance.
- Theoretical assumptions: Authors acknowledged strong/idealized assumptions; the theory remains only weakly tied to practical behavior.

**Reviewer Scores:**

- Reviewer **G6uc** (score 4): likely remains 4 given the still-missing baselines and fairness issues; could move to ~5 only if the requested baseline additions and controlled FT comparisons were completed and the paper cleaned up.
- Reviewer **4dsu** (score 6): likely remains 6; rebuttal addressed some points by pointing to cost/sensitivity analyses, but the pruning-side-effect measurement is still incomplete.
- Reviewer **oiCK** (score 6): likely remains 6 in score but would expect substantial proofreading and clearer empirical support for practical cost/scalability and the realism of theoretical assumptions.
- Reviewer **7U7d** (score 4): likely remains 4 unless the presentation is thoroughly fixed and the key missing baselines/comparisons (and RECIPE-style relation) are concretely addressed with experiments.

---

### Decision · Program_Chairs · 2026-01-26

Reject